# Pericyte-mediated constriction of renal capillaries evokes no-reflow and kidney injury following ischaemia

**Felipe Freitas\*, David Attwell\***

Department of Neuroscience, Physiology and Pharmacology, University College London, London, United Kingdom

**Abstract** Acute kidney injury is common, with ~13 million cases and 1.7 million deaths/year worldwide. A major cause is renal ischaemia, typically following cardiac surgery, renal transplant or severe haemorrhage. We examined the cause of the sustained reduction in renal blood flow ('no-reflow'), which exacerbates kidney injury even after an initial cause of compromised blood supply is removed. Adult male Sprague-Dawley rats, or NG2-dsRed male mice were used in this study. After 60 min kidney ischaemia and 30–60 min reperfusion, renal blood flow remained reduced, especially in the medulla, and kidney tubule damage was detected as Kim-1 expression. Constriction of the medullary descending vasa recta and cortical peritubular capillaries occurred near pericyte somata, and led to capillary blockages, yet glomerular arterioles and perfusion were unaffected, implying that the long-lasting decrease of renal blood flow contributing to kidney damage was generated by pericytes. Blocking Rho kinase to decrease pericyte contractility from the start of reperfusion increased the post-ischaemic diameter of the descending vasa recta capillaries at pericytes, reduced the percentage of capillaries that remained blocked, increased medullary blood flow and reduced kidney injury. Thus, post-ischaemic renal no-reflow, contributing to acute kidney injury, reflects pericytes constricting the descending vasa recta and peritubular capillaries. Pericytes are therefore an important therapeutic target for treating acute kidney injury.

## Editor's evaluation

This paper identifies constriction of capillary pericytes as the underlying cause of post-ischemia renal no-reflow conditions, which contribute to kidney injury in a variety of settings, including cardiac surgery, renal transplantation and severe haemorrhage. As such, it should be of considerable interest to clinicians, but also to basic researchers in vascular biology, physiology and related fields. Data obtained strongly support a role for RhoA/Rho kinase in regulating the contractility of capillary pericytes.

## Introduction

The global burden of acute kidney injury is approximately 13 million cases a year (*Ponce and Balbi, 2016*). It is associated with a high mortality (1.7 million deaths per year, worldwide) (*Gameiro et al., 2018*; *Hoste et al., 2018*; *Mehta et al., 2016*), and COVID-19 has added to its incidence (*Ronco et al., 2020*). Renal ischaemia followed by reperfusion, which can occur after cardiac surgery, renal transplant, or severe haemorrhage, is the most common cause of acute kidney injury (*Lameire et al., 2006*; *Lameire and Vanholder, 2001*). Sustained renal blood flow reductions occur after ischaemia and reperfusion, both in experimental studies and in patients after kidney transplantation (*Cristol et al., 1996*; *Nijveldt et al., 2001*; *Ramaswamy et al., 2002*). Following short periods of ischaemia,

**\*For correspondence:**
f.freitas@ucl.ac.uk (FF);
d.attwell@ucl.ac.uk (DA)

**Competing interest:** The authors declare that no competing interests exist.

**Figure 1.** Ischaemia and reperfusion lead to cortical and medullary no-reflow. (**a, b**) Ischaemia and reperfusion (I/R) evoked changes of blood flow (measured by laser Doppler) in the rat renal (**a**) medulla (n = 4 animals) and (**b**) cortex (n = 10 animals). CONT indicates blood flow on the contralateral (non-ischaemic) side. Traces labelled +HF show the effect on recovery of perfusion of administering the Rho kinase inhibitor hydroxyfasudil (HF) immediately on reperfusion (I/*R* + HF) (n = 4 animals). (**c–e**) Top: low power views of kidney slices after perfusion in vivo with FITC-albumin gelatin, from (**c**) control (contralateral) kidney, (**d**) a kidney after ischaemia and 30 min reperfusion, and (**e**) a kidney 30 min after treatment with HF on reperfusion **Bottom:** regions of interest (ROIs) are shown in red and blue for the cortex and medulla. (**f**) Medullary perfusion (assessed in slices of fixed kidney as the total intensity of FITC-albumin summed over the ROIs) was reduced after 30 min of post-ischaemic reperfusion (51 stacks, 6 animals) by ~50% compared with control kidneys (52 stacks, 7 animals). Treatment with HF increased medullary perfusion 2.3-fold at this time compared with non-treated ischaemic kidneys (20 stacks, 4 animals). (**g**) Cortex perfusion (assessed as in c-e) after 30 min of reperfusion after ischaemia was reduced by ~23.5% compared with control kidneys. Treatment with HF (I/*R* + HF) increased cortex perfusion by 25% at this time compared with non-treated ischaemic kidneys (I/R). Data are mean ± s.e.m. *P* values are corrected for multiple comparisons. Statistical tests used the number of animals as the N value (not the stack number).

The online version of this article includes the following source data and figure supplement(s) for figure 1:

**Source data 1.** Ischaemia and reperfusion lead to cortical and medullary no-reflow.

**Figure supplement 1.** Ischaemia (I/R) evoked changes of blood flow measured by laser Doppler in the rat renal medulla and cortex.

**Figure supplement 1—source data 1.** Ischaemia (I/R) evoked changes of blood flow measured by laser Doppler in the rat renal medulla and cortex.

blood flow to the renal cortex largely recovers following reperfusion, but medullary blood flow remains reduced for a prolonged period, especially in the hypoxia-sensitive outer medulla (the organisation of kidney areas and vasculature is shown in our summary Figure 8 below). Medullary no-reflow is a critical event for amplifying renal tissue injury following reperfusion (*Conesa et al., 2001*; *Olof et al., 1991*; *Regner et al., 2009*).

Renal no-reflow has been attributed to various causes, including impaired erythrocyte movement and leukocyte accumulation in renal capillaries, as well as increased intratubular pressure (*Bonventre and Weinberg, 2003*; *Sutton et al., 2002*; *Wei et al., 2017*; *Yamamoto et al., 2002*). However, after years of investigation, no effective treatment is available, even though no-reflow predicts a worse prognosis after kidney ischaemia. We therefore investigated an alternative possible cause of no-re-flow, that is ischaemia-evoked contraction of pericytes that regulate capillary diameter, which might reduce renal blood flow and physically trap red blood cells. Indeed, in the brain and heart contractile pericytes on capillaries play a key role in reducing blood flow after ischaemia (*Hall et al., 2014*; *O'Farrell et al., 2017*; *Yemisci et al., 2009*) because capillaries remain constricted by pericytes even when blood flow is restored to upstream arterioles. In the retina it has been shown that this capillary constriction is mediated by α-smooth muscle actin (α-SMA) based actomyosin-mediated contraction of capillary pericytes (*Alarcon-Martinez et al., 2019*). In the kidney, pericytes are associated with the cortical and medullary peritubular capillaries and the descending vasa recta. As in the retina, pericyte populations in the kidney, particularly those in the descending vasa recta, are associated with α-SMA expression and contractility (*Park et al., 1997*; *Shaw et al., 2018*). They play a key role in regulating renal medullary blood flow (*Crawford et al., 2012*; *Pallone and Silldorff, 2001*) which is a crucial variable for meeting the contradictory demands of preserving cortico-medullary osmotic gradients to allow water retention in the body, while maintaining adequate oxygen and nutrient delivery. This raises the question of whether pericytes also play a role in generating renal no-reflow after ischaemia.

An important regulator of pericyte contractility is the Rho kinase pathway (*Durham et al., 2014*; *Kutcher et al., 2007*), which inhibits myosin phosphatase, thus increasing phosphorylation of myosin light chain (MLC) and increasing contraction (*Kimura et al., 1996*; *Maeda et al., 2003*). Overactivity of Rho kinase may play a key role in hypertension and diabetes, as well as in kidney ischaemia (*Jahani et al., 2018*; *Kushiyama et al., 2013*; *Peng et al., 2008*; *Soga et al., 2011*; *Versteilen et al., 2006*). Rho kinase may also regulate pericyte contractility by modulating actin polymerisation (*Kureli et al., 2020*; *Kutcher and Herman, 2009*; *Maekawa et al., 1999*; *Zhang et al., 2018a*). In ischaemia, an important pathway by which Rho kinase inhibits myosin phosphatase is via inactivation of endothelial nitric oxide synthase (eNOS) (*Versteilen et al., 2006*), thus reducing production of nitric oxide (NO). NO acts on guanylate cyclase to raise the concentration of cyclic GMP, which increases MLC phosphatase activity and thus decreases contraction, so inhibiting eNOS will increase MLC phosphorylation and contraction. Thus, both the direct effect of Rho kinase (*Kimura et al., 1996*; *Maeda et al., 2003*) and its actions on eNOS (*Versteilen et al., 2006*) converge to promote MLC phosphorylation and contraction. Rho kinase is an important effector of vasoconstrictors such as endothelin-1 (*Prakash et al., 2008*; *Wilhelm et al., 1999*; *Yamamoto et al., 2000*) and angiotensin II (*Rupérez et al., 2005*), but its effects on pericytes are under-studied, although it may control their contractility (*Durham et al., 2014*; *Hartmann et al., 2021*; *Homma et al., 2014*; *Kutcher et al., 2007*; *Pearson et al., 2013*).

Few studies have investigated how ischaemia affects renal pericytes (*Kwon et al., 2008*; *McCurley et al., 2017*; *Zhang et al., 2018b*), and whether pericytes contribute to renal no-reflow. However, peritubular pericytes are damaged in cortical tissue of cadaveric renal allografts following ischaemia-reperfusion (*Kwon et al., 2008*), suggesting that renal blood flow control may be disrupted after ischaemia by pericyte dysfunction. Here, we show that pericyte-mediated capillary constriction, especially of the descending vasa recta, makes a crucial contribution to no-reflow following renal ischaemia and reperfusion. We further show that targeting pericyte-mediated constriction pharmacologically can reduce ischaemia-evoked acute kidney injury.

## Results

### No-reflow after renal ischaemia and reperfusion

Adult male Sprague-Dawley rats (P40-50), or NG2-dsRed male mice (P100-120) were used in this study. We used a combination of laser Doppler perfusion measurements, low magnification imaging of blood volume, and high magnification imaging that resolved individual capillaries, to assess the magnitude and cause of changes of renal perfusion after ischaemia. Ischaemia for 1 hr decreased perfusion of the renal medulla and cortex by ~90% (both $P < 0.0001$ vs. control; assessed with laser Doppler: *Figure 1a and b*). After 30 min reperfusion, blood flow recovered to 49% of control (significantly reduced, $P = 0.005$, *Figure 1a*) in the medulla, but to 75% in the cortex ($P = 0.047$, *Figure 1b*; *Regner et al., 2009*). Perfusion was stable in the contralateral kidney throughout (*Figure 1a and b*). After 60 min reperfusion, medullary perfusion remained compromised at 40% of the control level ($P = 0.017$, *Figure 1—figure supplement 1a*), but cortical perfusion had fully recovered (to ~20% above the control value, not significant, $P = 0.092$, *Figure 1—figure supplement 1b*). Despite this flow recovery, we show below that peritubular capillaries in the cortex can become blocked after ischaemia.

After ischaemia and reperfusion in vivo, assessing the volume of perfused vessels in fixed kidney slices, as the summed FITC-albumin intensity over ROIs, also demonstrated that renal ischaemia and reperfusion led to no-reflow in the medulla compared with the non-ischaemic kidney's medulla (the perfusing blood volume was reduced by ~50%, $P = 0.002$; *Figure 1c, d and f*). Microscopic analysis resolving individual capillaries showed that this blood volume reduction was associated with a large reduction in capillary perfusion (*Figure 2*). The total perfused capillary length in 100 μm deep confocal z-stacks (frame size 640.17 × 640.17 μm) was reduced by 35% (contralateral control 14689 ± 3477 μm vs. ischaemia 9527 ±1183 μm, $P = 0.038$), the number of perfused capillary segments was reduced by 54% (control 530 ± 82 vs. ischaemia 244 ± 30, $P = 0.03$), and the overall perfused microvascular volume fraction was reduced by 51% (control 0.116 ± 0.006 vs. ischaemia 0.057 ± 0.006, $P = 0.003$; *Figure 2e–g*).

In the cortex, perfusion was reduced less than in the medulla after ischaemia and reperfusion, that is by 23.5% compared with non-ischaemic kidneys ($P = 0.0075$, *Figure 1c, d and g*). Furthermore, although a small percentage of afferent and efferent arterioles, and glomeruli, were not perfused in control conditions, this percentage did not increase significantly after ischaemia (*Figure 3a, b and g*), and the arterioles' diameter was not reduced compared with those in non-ischaemic kidneys (*Figure 3a, b, h and i*). Similarly, it has been reported that upstream arteries are not constricted after ischaemia (*Yamamoto et al., 2002*). In contrast, the total perfused peritubular capillary length in the 100 μm deep z-stacks (control 16441 ± 1577 μm vs. ischaemia 5411 ±2735 μm, reduced by 67%, $P = 0.03$), the number of perfused capillary segments (control 550 ± 32 μm vs. ischaemia 349 ±54, reduced by 36.5%, $P = 0.01$) and the overall perfused peritubular capillary volume fraction (control 0.12 ± 0.01 vs. ischaemia 0.06 ± 0.02, reduced by 50%, $P = 0.01$) were greatly reduced in the cortex when compared with non-ischaemic kidneys (*Figure 3d–f*). Thus, the effect of ischaemia and reperfusion is predominantly on the microvasculature, that is the peritubular cortical capillaries and the vasa recta, rather than on arteriolar segments of the kidney circulation. The Rho kinase inhibition data shown in *Figure 3* are discussed below.

### Pericytes constrict descending vasa recta after ischaemia and reperfusion

Higher magnification images demonstrated that, in control kidneys, only 9.7% of the descending vasa recta (DVR) capillaries were blocked (*Figures 2b and 4d*), that is were not perfused by FITC-albumin (*Figures 2c and 4a–d*). However, after ischaemia and 30 mins reperfusion, 78% of the DVR capillaries were blocked (*Figures 2c and 4a–d*). Some capillaries were fully perfused and some completely unperfused throughout the area assessed, whereas some exhibited an abrupt cessation of blood flow with a decrease of FITC-albumin intensity over a few microns (*Figures 2c and 4a–c*). At block sites, the diameter of the FITC-albumin lumenal labelling at the final position blood reached was significantly lower in ischaemic DVR capillaries compared with that at the much smaller number of block sites in non-ischaemic controls (control 6.5 ± 0.3 μm vs. ischaemia 3.5 ± 0.4 μm; $P = 0.039$, *Figure 4e*). Thus, an ischaemia-induced constriction of the DVR promotes blockage, which persists even after reperfusion.

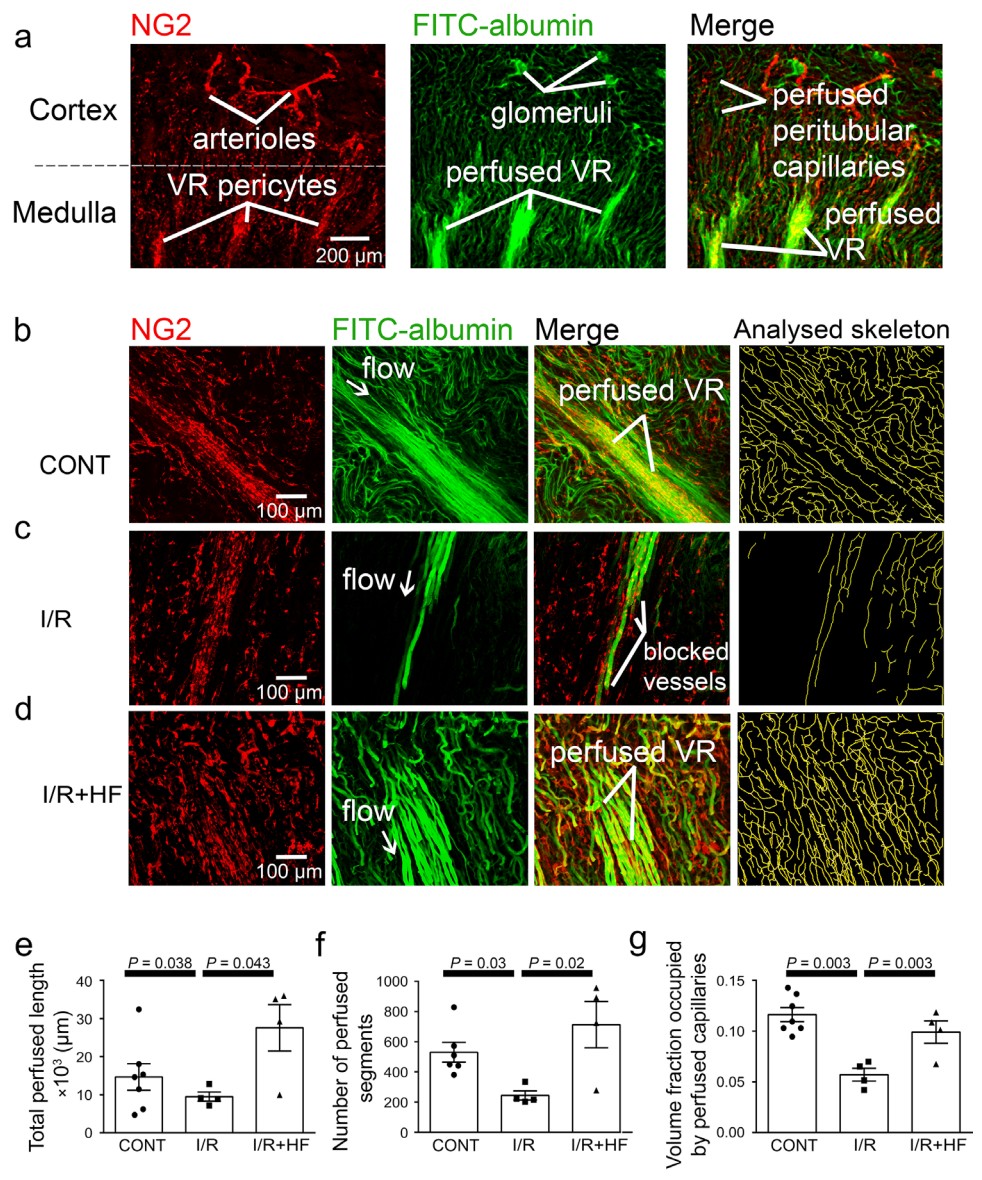

**Figure 2.** Ischaemia and reperfusion reduce medullary microvascular perfusion. (**a**) Representative images of slices after perfusion with FITC-albumin gelatin, showing the rat kidney microcirculation in 100 μm deep confocal z-stacks. Images depict renal cortical arterioles, the glomeruli and peritubular capillaries, as well as the vasa recta capillaries (VR) that supply blood to the renal medulla. (**b–d**) Representative images of the medullary microcirculation: (**b**) in control conditions (CONT), (**c**) after ischaemia and 30 min reperfusion (I/R), and (**d**) after ischaemia and reperfusion for 30 min with hydroxyfasudil (HF) applied during reperfusion (I/*R* + HF). Images show NG2-labelling of pericytes (red), FITC-albumin labeling (green) of vessels that are perfused, a merge of the NG2 and FITC-albumin images, and the analysed skeleton (yellow) of the perfused microvessels. (**e–g**) After ischaemia and reperfusion (12 stacks, 4 animals), the total perfused capillary length (**e**), the number of perfused capillary segments (**f**) and the overall volume fraction of vessels perfused (**g**) in 100 μm deep confocal z-stacks were reduced compared with control kidneys (14 stacks, 6–7 animals), and treatment with hydroxyfasudil immediately after reperfusion (10 stacks, 4 animals) increased all of these parameters. Data are mean ± s.e.m. *P* values are corrected for multiple comparisons. Statistical tests used the number of animals as the N value (not the stack number).

The online version of this article includes the following source data for figure 2:

**Source data 1.** Ischaemia and reperfusion reduce medullary microvascular perfusion.

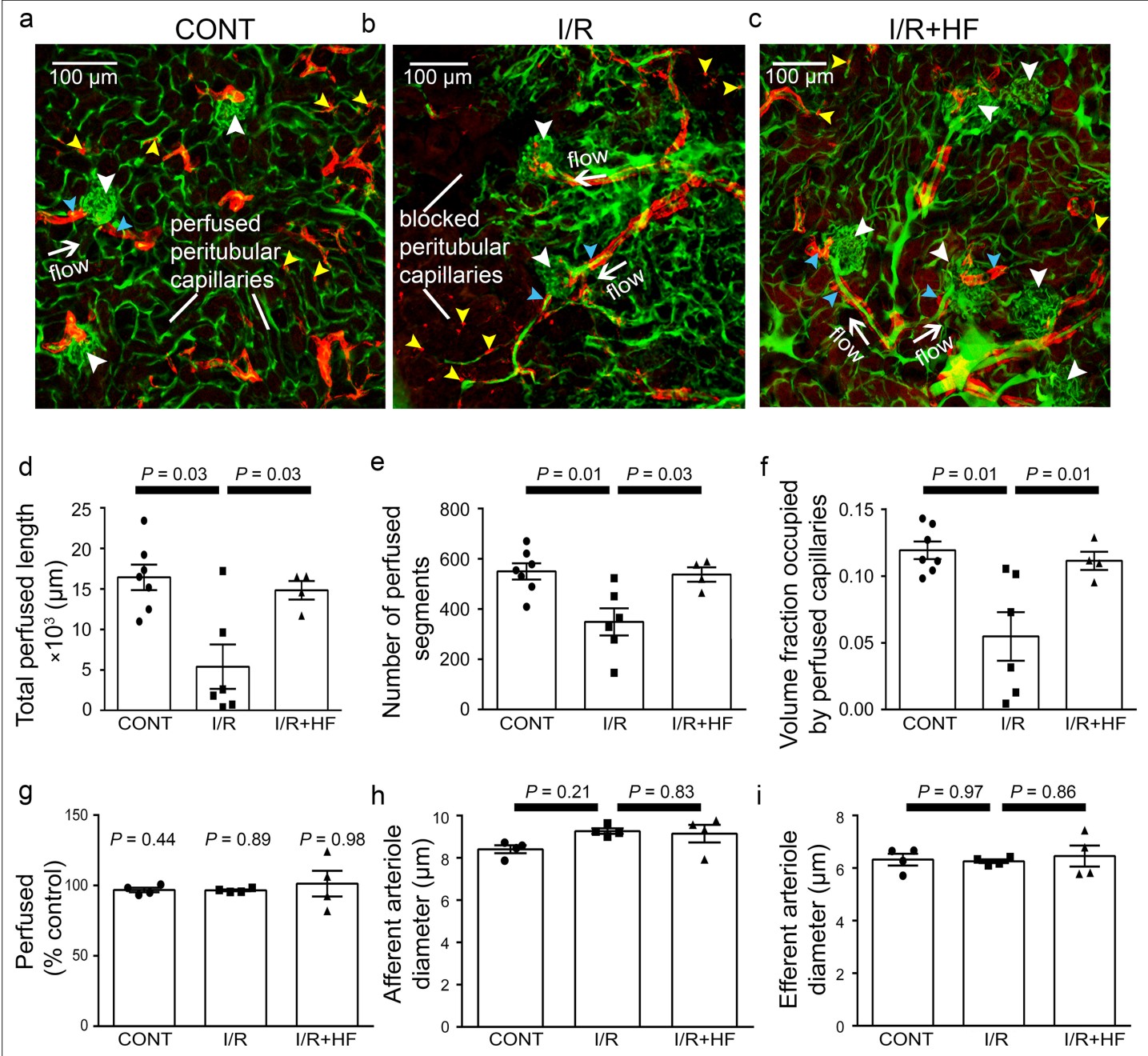

**Figure 3.** Ischaemia and reperfusion of renal cortex evoke no-reflow in capillaries but not arterioles. (**a–c**) Representative images of rat renal cortex slices containing arterioles, glomeruli and peritubular capillaries, after perfusion with FITC-albumin gelatin: (**a**) for control kidneys (CONT), (**b**) after ischaemia and reperfusion (I/R), and (**c**) after ischaemia with hydroxyfasudil (I/R + HF). NG2-labelling (red) is seen of arterioles (blue arrowheads) and pericytes (yellow arrowheads), while FITC-albumin labelling (green) shows vessels that are perfused. (**d-f**) After ischaemia and reperfusion (I/R) (12 stacks, 6 animals), the total perfused capillary length (**d**), the number of perfused segments (**e**), and the overall perfused microvascular volume fraction (**f**) were reduced compared with control kidneys (CONT) (14 stacks, 7 animals), and treatment with hydroxyfasudil immediately after reperfusion (I/R + HF) (10 stacks, 4 animals) increased cortical microvascular perfusion compared with non-treated ischaemic kidneys. (**g**) Percentage of afferent and efferent arterioles (blue arrowheads in a-c), and of glomeruli (white arrowheads), perfused after ischaemia, compared with control conditions. (**h–i**) Diameters of perfused (**h**) afferent and (**i**) efferent arterioles in the renal cortex for the three experimental conditions (15 arterioles, 4 animals for each group). Data are mean ± s.e.m. *P* values are corrected for multiple comparisons.Statistical tests used the number of animals as the N value (not the stack number).

The online version of this article includes the following source data for figure 3:

**Source data 1.** Ischaemia and reperfusion of renal cortex evoke no-reflow in capillaries but not arterioles.

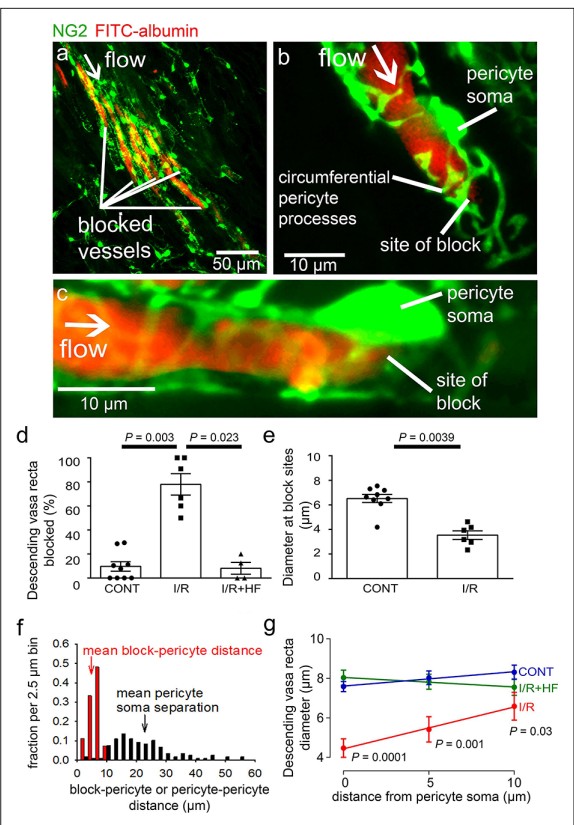

**Figure 4.** Descending vasa recta are constricted by pericytes after ischaemia. (**a**) Descending vasa recta (DVR) in slices of rat renal medulla after perfusion with FITC-albumin gelatin (re-coloured red), and labelled for pericytes with antibody to the proteoglycan NG2 (green); FITC-albumin labelling shows perfused and blocked vessels. White arrow indicates flow direction; white lines indicate blocked vessels. (**b–c**) Representative images showing DVR capillaries blocked near pericyte somata. NG2-labelling of pericytes shows pericyte processes presumed to be constricting vessels at block site. (**d**) Percentage of DVR capillaries blocked in the renal medulla in control conditions (CONT) (127 capillaries, 12 stacks, 9 animals), after ischaemia and reperfusion (I/R) (77 capillaries, 10 stacks, 6 animals), and after ischaemia with hydroxyfasudil present in the reperfusion period (IR + HF) (60 capillaries, 8 stacks, 4 animals). Statistical tests used number of animals as the N value. (**e**) Diameter at block sites. (**f**) Probability distribution per 2.5 µm bin of distance from blockage to nearest pericyte soma after ischaemia and reperfusion (for 27 block sites), and of the distance between adjacent pericytes on DVR capillaries (for 118 pericyte pairs). (**g**) DVR diameter versus distance from pericyte somata (10 µm is approximately half the separation between pericytes) in the same three conditions as d (number of pericytes was 31, 20, and 17 respectively). *P* values by each point are from t-tests. Slope of the best-fit ISCH regression line is significantly greater than zero (*P* = 0.039) while that of the CONT line is not (*P* = 0.084). Data are mean ± s.e.m.

The online version of this article includes the following source data and figure supplement(s) for figure 4:

**Source data 1.** Descending vasa recta are constricted by pericytes after ischaemia.

**Figure supplement 1.** The effect of renal ischaemia and reperfusion on red blood cell trapping and endothelial glycocalyx integrity in the descending vasa recta.

**Figure supplement 1—source data 1.** The effect of renal ischaemia and reperfusion on red blood cell trapping and endothelial glycocalyx integrity in the descending vasa recta.

**Figure supplement 1—source data 2.** The effect of renal ischaemia and reperfusion on red blood cell trapping and endothelial glycocalyx integrity in the descending vasa recta.

**Figure supplement 2.** Morphology of renal pericytes.

Erythrocyte protein glycophorin A was labelled to assess if red blood cells were trapped at capillary regions of reduced diameter. Red blood cells were associated with only a small percentage of blockage sites in ischaemic kidneys (5.8% of 85 blockages in 137 vessels from two animals), and even where red blood cells were near the capillary blockages they did not always block blood flow because FITC-albumin could pass the red blood cells (*Figure 4—figure supplement 1a, b*).

In the brain (*Hall et al., 2014*; *Yemisci et al., 2009*) and heart (*O'Farrell et al., 2017*), post-ischaemic capillary constriction reflects pericyte contraction, which occurs near pericyte somata where circumferential processes originate (*Nortley et al., 2019*). From NG2 labelling, we observed that many DVR blockages were close to pericyte somata, or near to pericyte circumferential processes connected to the soma (*Figure 4b–c*), suggesting that contraction of these juxta-somatic processes evoked capillary block. We measured the distance of 27 blockages to the nearest pericyte soma. The probability distribution of this distance is compared with that of the inter-pericyte distance in *Figure 4f* (if blocks did not depend on pericytes, the probability distribution of the blockage-pericyte distance would be constant until half the distance between pericytes). The mean blockage-pericyte distance was 4.87 ± 0.33 µm after ischaemia and reperfusion, which is less than a quarter of the distance between DVR pericytes (22.85 ± 0.93 µm, from 118 pericyte pairs). Thus, these data are consistent with pericyte constriction generating the DVR blockages.

In control conditions, the few blockages occurring were mainly in regions where the inter-pericyte distance was larger. The mean distance from a blockage to the nearest pericyte soma was also larger (14.98 ± 1.36 µm, $P < 0.0001$ compared to post-ischaemia), suggesting a different block mechanism in control conditions.

To assess pericyte-mediated DVR constriction further, we measured the FITC-albumin labelled lumen diameter at 5 µm intervals upstream of pericyte somata (upstream so there was FITC-albumin in the vessel: *Figure 4g*). After ischaemia and reperfusion, the diameter was significantly reduced (by 41%, $P = 0.0001$) near the pericyte somata compared with non-ischaemic kidneys, but less reduced further from the somata. The diameter significantly increased with distance from the somata after ischaemia and reperfusion ($P = 0.039$ comparing the slope of the best-fit ischaemia regression line with zero) but not in control conditions ($P = 0.084$), implying constriction preferentially near the pericyte somata (*Figure 4g*) and identifying pericytes as the origin of the diameter reduction. Such constrictions will reduce blood flow directly by increasing the vascular resistance, and may also lead to blood cells becoming trapped at the regions of narrowed diameter, thus occluding the vessel and further reducing blood flow.

We assessed whether the endothelial glycocalyx (eGCX) contributed to DVR blockages. Labelling showed that eGCX is fairly uniformly present along capillaries, and this was not altered after ischaemia (*Figure 4—figure supplement 1f-g*). There was no correlation between eGCX intensity and capillary diameter in control or ischaemic conditions (*Figure 4—figure supplement 1h*). Thus, eGCX is not particularly associated with pericytes (*Figure 4—figure supplement 1f*), so the co-location of diameter reduction and blockages with pericyte somata presumably reflects pericyte process contraction rather than obstruction by eGCX.

## Pericytes constrict peritubular cortical capillaries in vivo after ischaemia and reperfusion

Two-photon microscopy in vivo, of mice expressing dsRed in pericytes, revealed peritubular cortical pericytes constricting and blocking capillaries after ischaemia and reperfusion (*Figure 5a–c*). This reduced the mean capillary diameter (averaged over all positions measured) from 10.8 ± 0.2 to 8.1 ± 0.5 µm ($P < 0.0001$). To quantify whether ischaemia-evoked blockages occurred disproportionately close to pericytes, we measured the distance of 15 blockages to the nearest pericyte soma. This distance was 4.12 ± 0.39 µm, which is only 10% of the mean distance between peritubular cortical pericytes (41.3 ± 2.6 µm, from 103 pericyte pairs). A plot of capillary diameter versus distance from pericyte somata (*Figure 5d*) showed that ischaemia and reperfusion reduced the diameter by 40% at the somata (control 11.2 ± 0.5 vs. ischaemia 6.76 ± 1.05 µm, $P = 0.001$) with no significant effect on diameter far from the somata (control 10.3 ± 0.2 µm vs. ischaemia 9.6 ± 0.5 µm, $P = 0.115$). As in the medulla, the diameter increased significantly with distance from the pericyte somata after ischaemia ($P = 0.046$ comparing the slope of the best-fit regression line with zero) while in control conditions it did not (diameter decreased insignificantly with distance, $P = 0.10$). Thus, capillaries are constricted specifically near cortical pericytes.

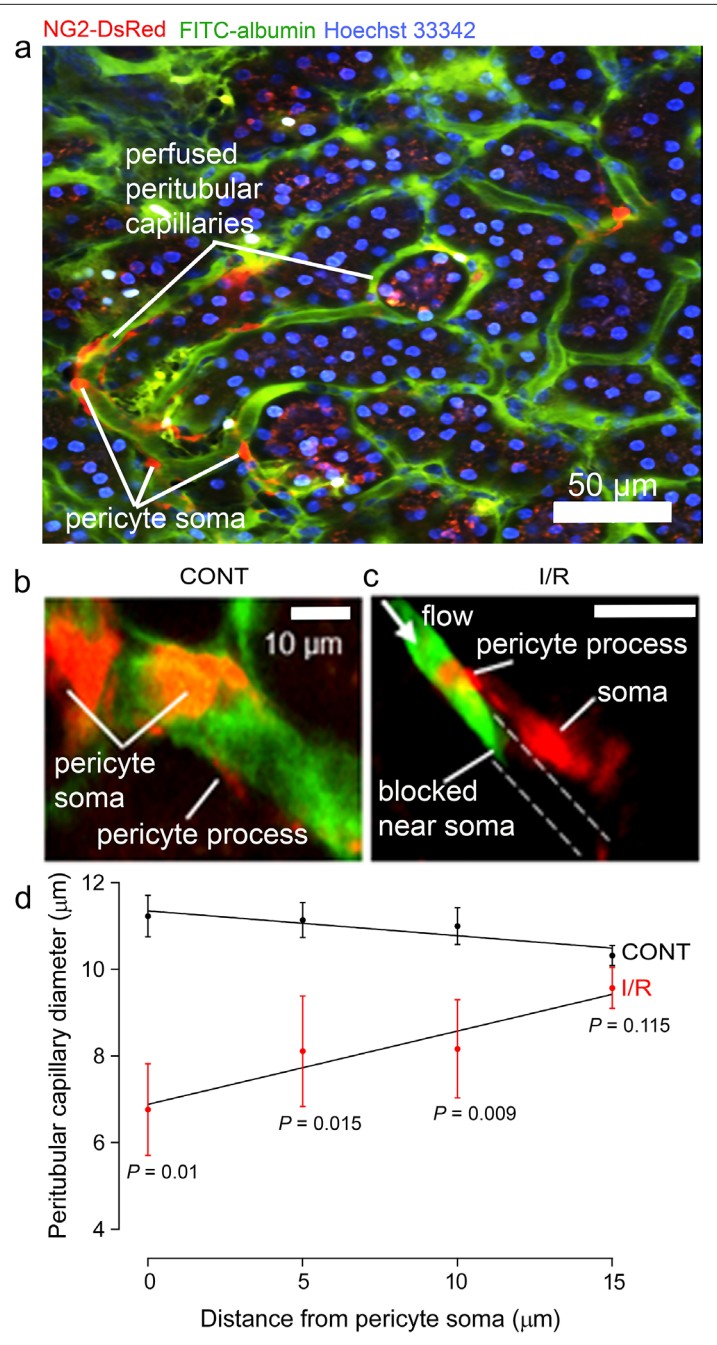

**Figure 5.** Pericytes constrict capillaries after renal ischaemia in vivo. (**a**) Overview two-photon in vivo imaging stack of the mouse renal cortex microcirculation, showing pericytes expressing NG2-DsRed (red), intraluminal FITC-albumin given intravenously (green), and Hoechst 33,342 labelling nuclei (blue, 1 mg/kg in 0.5 ml of sterile, isotonic saline was administered intravenously: *Dunn et al., 2018*). Images were acquired in a plane parallel to the cortical surface. (**b, c**) Higher magnification images showing apericyte on a cortical peritubular capillary in control conditions, and post-ischaemic capillary block (dashed lines show path of blocked vessel). (**d**) Capillary diameter versus distance from pericyte somata after ischaemia and reperfusion (I/R), and for control kidneys (CONT) (number of pericytes was 15 and 10 respectively from 10 stacks from three animals from each group). Slope of the best-fit ISCH regression line is significantly greater than zero (*P* = 0.046) while that of the CONT line is negative but not significantly different from zero (*P* = 0.10). Data are mean ± s.e.m. *P* values comparing data at each distance are corrected for multiple comparisons. Statistical tests used number of stacks as the N value.

The online version of this article includes the following source data for figure 5:

**Source data 1.** Pericytes constrict capillaries after renal ischaemia in vivo.

The fact that pericyte constriction of capillaries reduces blood flow more in the medulla than in the cortex (*Figure 1*) may at least partly reflect differences in pericyte number and morphology in these two regions. The mean distance between pericytes in the medulla (23 μm, see above) is roughly half that in the cortex (41 μm, see above). Furthermore, in general the morphology of pericytes differs in these two regions, with DVR pericytes showing many circumferential processes around the capillaries, while cortical pericytes exhibit mainly longitudinal processes running along the capillary with only a small number of circumferential processes (*Figure 4—figure supplement 2*). The small number of vessel branches in the medullary DVR implies that the class of pericyte associated with branch points that is found in the brain vasculature will be less common here.

## Rho kinase inhibition reduces pericyte constriction and no-reflow

The contractility of pericytes depends partly on Rho kinase activity (*Durham et al., 2014*; *Hirunpattarasilp et al., 2019*; *Homma et al., 2014*; *Kutcher et al., 2007*). The Rho kinase inhibitor, hydroxyfasudil (3 mg/kg; i.v.), applied at the time of reperfusion to mimic a possible therapeutic intervention, significantly inhibited the decrease of renal medullary perfusion seen after ischaemia-reperfusion (*Figure 1a and e–f*). In vivo, blood flow in the medulla (after 30 min reperfusion) was increased 3.8-fold compared to ischaemia without hydroxyfasudil ($P = 0.002$, *Figure 1a*). Hydroxyfasudil induced a faster recovery of medullary blood flow than BQ123 (0.5 mg/kg, i.v.), an endothelin-A receptor antagonist (*Figure 1—figure supplement 1c*), but both resulted in blood flow at 30 mins reperfusion that was not significantly different from the control value ($P = 0.8$ and $0.38$, respectively) and was significantly higher than the flow seen after ischaemia without either drug ($P = 0.01$ for both drugs). In contrast, the angiotensin II type 1 (AT1) receptor antagonist valsartan (1 mg/kg i.v.) speeded the initial post-ischaemic recovery of medullary blood flow, but did not return it to baseline by 30 min reperfusion (*Figure 1—figure supplement 1c*). In the cortex, blood flow recovery on reperfusion was speeded by hydroxyfasudil and, after 30 min of reperfusion, was increased 1.48-fold compared to ischaemia alone ($P = 0.02$, *Figure 1b*).These data suggest that, in the medulla especially, activation of Rho kinase (in part downstream of ischaemia-evoked activation of endothelin-A receptors [*Prakash et al., 2008*; *Wilhelm et al., 1999*; *Yamamoto et al., 2000*]) contributes to ischaemia-evoked pericyte-mediated capillary constriction.

Renal perfusion with post-ischaemic inhibition of Rho kinase was also assessedin slices of fixed kidney (see above). Treatment with hydroxyfasudil during post-ischaemic reperfusion prevented medullary no-reflow after ischaemia and reperfusion: the blood volume was increased 2.3-fold compared to ischaemia alone ($P = 0.003$, *Figure 1e–f*), so that it did not differ significantly from that in control kidney ($P = 0.47$). Hydroxyfasudil also increased ~2.9-fold the total perfused medullary capillary length ($P = 0.043$),~2.9 fold the number of perfused capillary segments ($P = 0.02$) and ~2-fold the perfused volume fraction ($P = 0.0031$) in medulla (*Figure 2d–g*). In the renal cortex, hydroxyfasudil given on reperfusion increased perfusion (blood volume)~1.25-fold ($P = 0.0098$; *Figure 1e and g*), and increased the total perfused length of capillaries, the number of perfused capillary segments and the blood volume fraction to values that were not significantly different from those in non-ischaemic kidneys (*Figure 3c–f*).

## Improvements of renal blood flow by hydroxyfasudil are via pericytes, not arterioles

Hydroxyfasudil might act on arteriolar smooth muscle or pericytes, or both. However, it had no effect on the diameter of afferent or efferent arterioles feeding and leaving the glomeruli (*Figure 3h and i*). In contrast, hydroxyfasudil reduced the constriction evoked at DVR pericyte somata by ischaemia and reperfusion, increasing the diameter from 4.5 ± 0.5 μm without hydroxyfasudil to 8.0 ± 0.4 μm with the drug ($P < 0.0001$) (*Figure 4g*), and reduced the percentage of DVR capillaries blocked from 78 ± 9% to 8 ± 5% ($P = 0.023$), both of which are not significantly different from the values in non-ischaemic kidneys (*Figure 4d and f*). Thus, ischaemia induces, and hydroxyfasudil decreases, medullary no-reflow by specifically acting on DVR capillary pericytes rather than on upstream arterioles.

## Rho kinase inhibition reduces myosin light chain phosphorylation after ischaemia

Rho kinase can inhibit, either directly or by inhibiting eNOS (*Riddick et al., 2008*; *Wang et al., 2009*; *Versteilen et al., 2006*), myosin light chain phosphatase (MLCP), thus increasing phosphorylation of myosin light chain (MLC) by myosin light chain kinase (MLCK) and increasing pericyte contraction, but it also has other functions. To investigate how Rho kinase inhibition has the effects described above, we labelled for phosphorylated MLC. After ischaemia and reperfusion, this was increased ~11-fold for medullary and five-fold for cortical pericytes ($P = 0.0001$ in both locations, *Figure 6a–j*). Hydroxy-fasudil treatment after reperfusion reduced this increase so that the labelling was not significantly different from that in control kidneys ($P = 0.95$ and $P = 0.56$, respectively; *Figure 6a–j*). Thus, if peri-cyte contraction is via conventional smooth muscle actomyosin, the reduced MLC phosphorylation could explain the pericyte relaxation and increased blood flow evoked by Rho kinase inhibition. The data of *Versteilen et al., 2006* suggest this is very largely mediated by inhibition of eNOS, which could be tested by quantifying the effect of eNOS block on the changes of MLC phosphorylation shown in *Figure 6*. Consistent with pericytes employing smooth muscle actomyosin, 56% of DVR pericytes near blockage sites labelled for the contractile protein α-SMA (*Figure 6k–n*; see also *Park et al., 1997*).

## Rho kinase inhibitor reduces reperfusion-induced acute kidney injury

Kidney injury molecule-1 (Kim-1) is a sensitive and early diagnostic indicator of renal injury in rodent kidney injury models (*Vaidya et al., 2010*), and in pathology is localised at high levels on the apical membrane of the proximal tubule where the tubule is most affected (*Amin et al., 2004*; *Ichimura et al., 1998*). Kim-1 levels in the proximal tubules were elevated 81-fold by ischaemia and reperfusion ($P = 0.0004$, *Figure 7a, b and d*), and treatment with hydroxyfasudil during reperfusion halved the Kim-1 labelling ($P = 0.03$, *Figure 7c and d*).

## Discussion

This paper demonstrates, for the first time, that the long-lasting decrease of renal blood flow that follows transient ischaemia is generated by pericyte-mediated constriction and block of the descending vasa recta and cortical peritubular capillaries, as schematised in the summary of *Figure 8*, and that this post-ischaemic no-reflow can be reduced pharmacologically. We found in vivo that sites of ischaemia-evoked medullary and cortical capillary block were associated with pericyte locations. Furthermore, after ischaemia and reperfusion, the diameters of descending vasa recta and peritu-bular capillaries were reduced specifically near pericyte somata, which extend contractile circumfer-ential processes around the capillaries. In contrast, cortical arteriole diameters were not reduced and glomeruli remained perfused. The fact that capillary diameters are reduced specifically near pericyte somata establishes that this is due to a contraction of the circumferential processes of pericytes, and not (for example) due to a decrease in overall perfusion pressure (which would also reduce the diameter of capillaries away from pericyte somata). Together, these data establish pericyte-mediated capillary constriction as a major therapeutic target for treating post-ischaemic renal no-reflow.

Pericyte-mediated constriction of renal capillaries may reflect reduced $Ca^{2+}$ pumping in ischaemia, raising $[Ca^{2+}]_i$ which activates contraction, as for CNS pericytes (*Hall et al., 2014*). Constriction may also partly reflect a release of angiotensin II (*Allred et al., 2000*; *Boer et al., 1997*; *da Silveira et al., 2010*; *Miyata et al., 1999*; *Sanchez-Pozos et al., 2012*; *Zhang et al., 2004*) and endothelin-1 (*Afyouni et al., 2015*; *Jones et al., 2020*; *Sanchez-Pozos et al., 2012*) which raise $[Ca^{2+}]_i$ and Rho kinase activity (*Lee et al., 2014*; *Shimokawa and Rashid, 2007*), since we found that blocking endo-thelin-A receptors and, to a lesser extent, angiotensin II receptors improved post-ischaemic renal blood flow. Consistent with this, it has been demonstrated that vasoconstricting endothelin-A (*Craw-ford et al., 2012*; *Wendel et al., 2006*) and angiotensin II type 1 (AT1) (*Crawford et al., 2012*; *Miyata et al., 1999*; *Terada et al., 1993*) receptors are located on pericytes along the descending vasa recta and regulate contractility at pericyte sites (*Crawford et al., 2012*). Additionally, endothelin-1 and angiotensin II evoke potent vasoconstriction of the descending vasa recta mainly through endothe-lin-A (*Silldorff et al., 1995*) and angiotensin II type 1 (AT1) (*Rhinehart et al., 2003*) receptors.

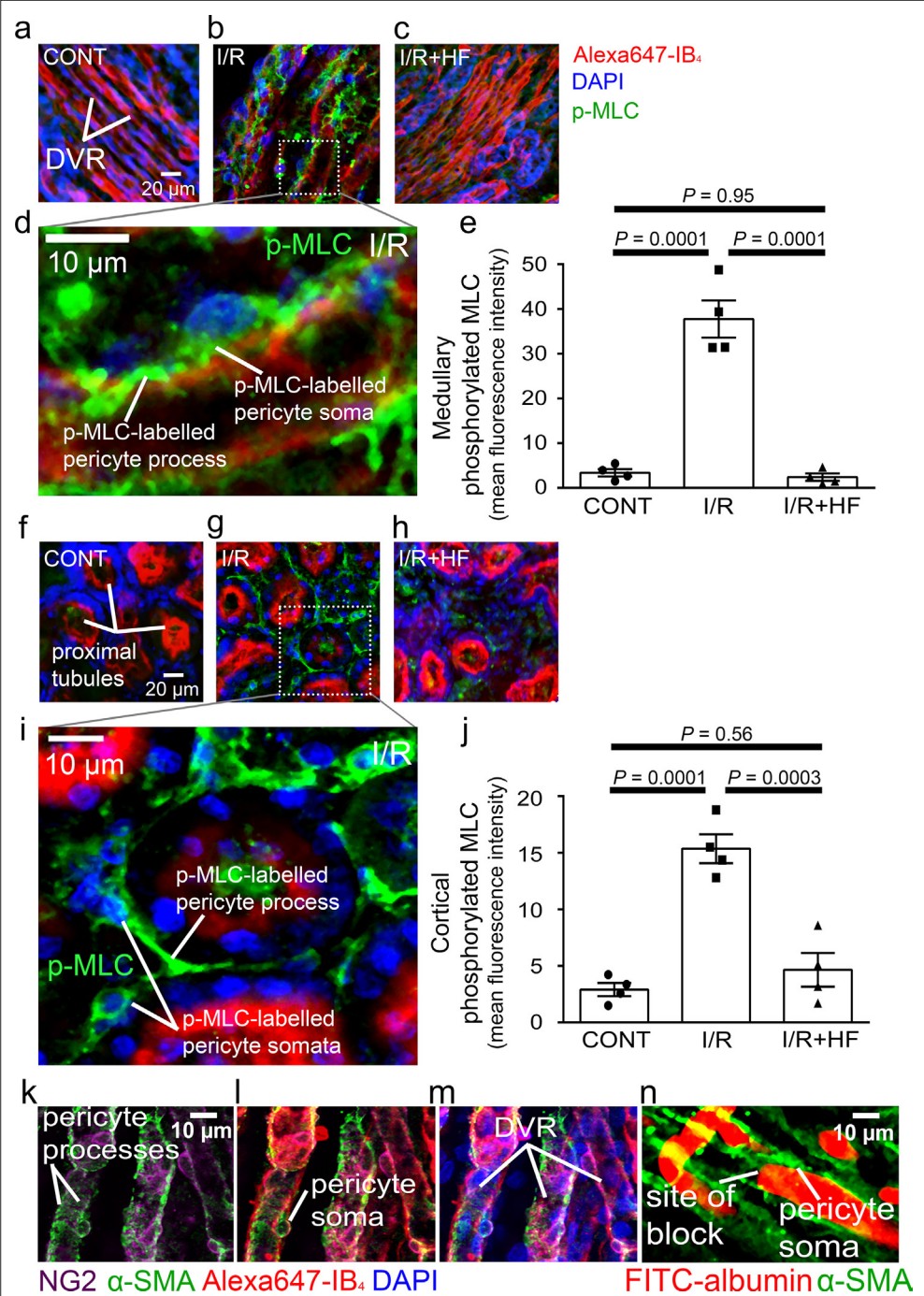

**Figure 6.** Pericyte contraction is mediated by α-SMA and regulated by Rho kinase. Representative images of the rat renal medulla containing descending vasa recta (DVR) pericytes (**a–d**) and cortical peritubular capillary pericytes (**f–i**), labelled with antibody to phosphorylated myosin light chain (p-MLC, green), Alexa Fluor 647-isolectin $B_4$ which labels kidney tubules and pericytes (red), and DAPI which labels nuclei (blue). Labelling is shown for kidneys in control conditions (CONT) (**a, f**), after ischaemia and reperfusion (I/R) (**b, d, g, i**), and after ischaemia with hydroxyfasudil present during reperfusion (I/R + HF) (**c, h**). (**e, j**) Cortical (**e**) and medullary (**j**) p-MLC levels in pericytes for the three experimental conditions (10 stacks, 4 animals for each group). (**k–m**) DVR pericytes labelled for NG2 (purple), α-SMA (green), Alexa647-isolectin B4 (red) and DAPI (blue). (**n**) DVR blockage-associated pericyte labelled for α-SMA. Statistical tests used the numbers of animals for N values (not the stack number). Data are mean ± s.e.m. *P* values are corrected for multiple comparisons.

The online version of this article includes the following source data for figure 6:

**Source data 1.** Pericyte contraction is mediated by α-SMA and regulated by Rho kinase.

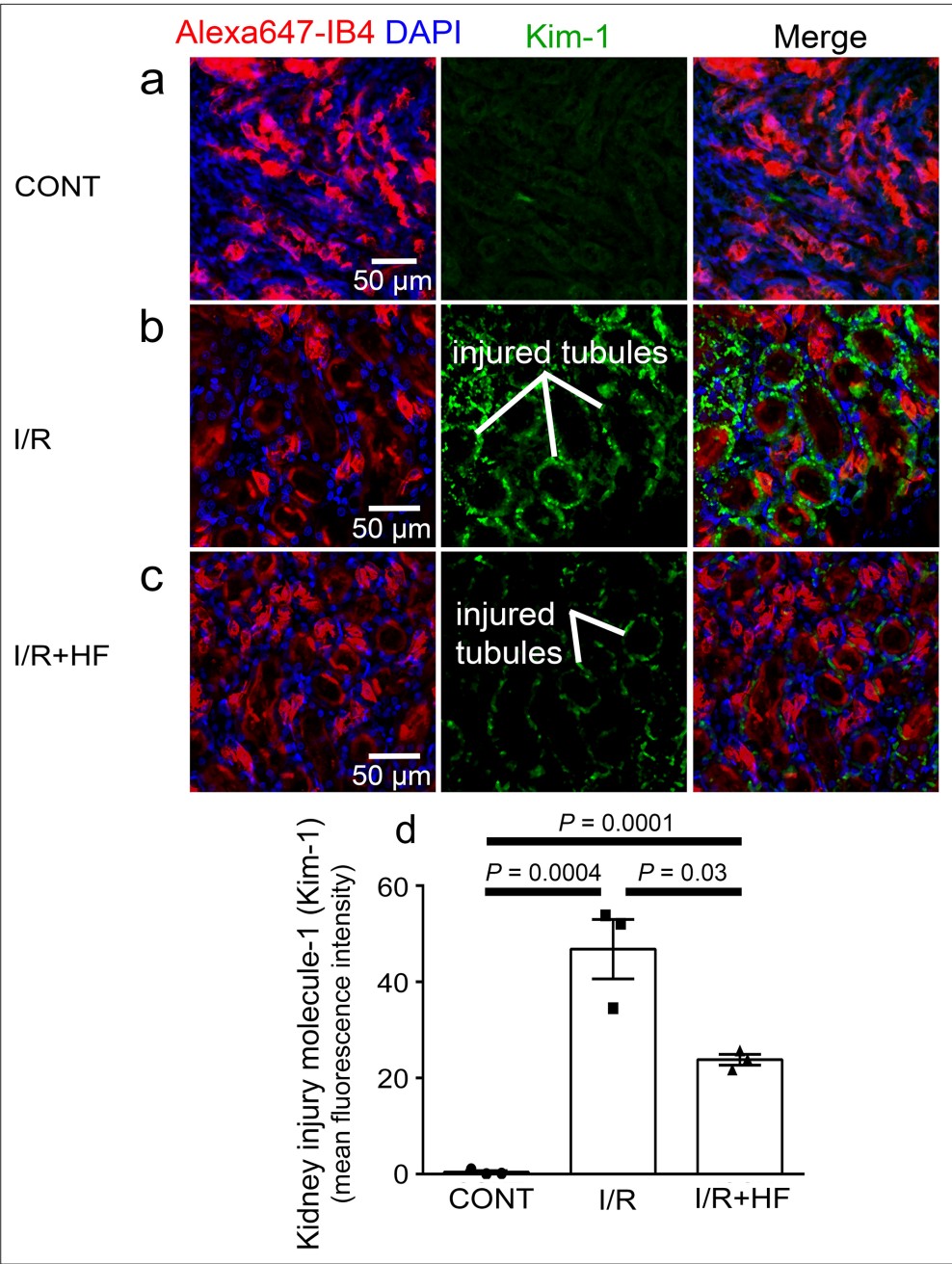

**Figure 7.** Rho kinase inhibition reduces kidney injury induced by ischaemia and reperfusion. (**a–c**) Images of the rat renal cortex containing proximal tubules, showing isolectin B₄ labelling kidney tubules (red), DAPI labelling nuclei (blue), and kidney injury molecule-1 (Kim-1) labelling as an injury marker (white lines indicate examples of injured tubules labelled in green), for control conditions (CONT) (**a**), after ischaemia and reperfusion (I/R) (**b**), and after ischaemia with hydroxyfasudil present during reperfusion (I/*R* + HF) (**c**).(**d**) Kim-1 levels for the three experimental conditions (six stacks, 3 animals for each group). Data are mean ± s.e.m. *P* values are corrected for multiple comparisons. Statistical tests used the number of animals as the N value (not the stack number).

The online version of this article includes the following source data for figure 7:

**Source data 1.** Rho kinase inhibition reduces kidney injury induced by ischaemia and reperfusion.

It has long been known that some pericyte populations in the kidney, especially those in the descending vasa recta, express α-SMA and regulate capillary blood flow (***Park et al., 1997***; ***Shaw et al., 2018***), presumably via actomyosin-based contractility. A potentially important physiological role for the presence of α-SMA in the descending vasa recta pericytes is the ability of these

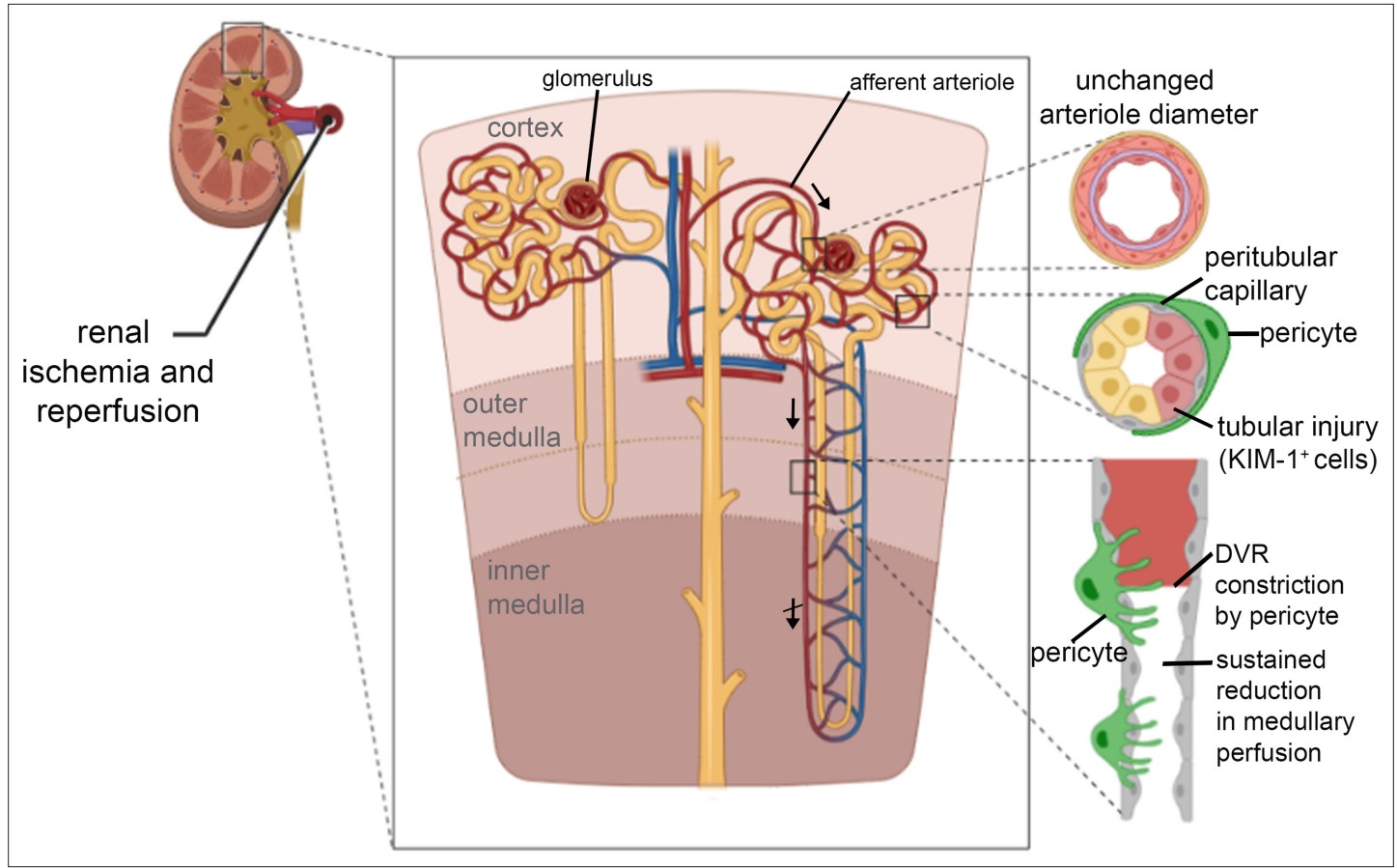

**Figure 8.** Schematic diagram of loci of blood flow reductions after renal ischaemia and reperfusion. The afferent arteriole feeding the glomerulus (top arrow) and the efferent arteriole leaving the glomerulus are little affected by ischaemia and reperfusion. In contrast, pericytes on peritubular capillaries and the descending vasa recta (upper descending arrow) constrict the capillaries, reducing blood flow and causing blockages as schematised at the lower right, and indicated by the crossed lower descending arrow signifying impaired DVR flow. The resulting ischaemia leads to kidney damage detectable by Kim-1 labelling. Hydroxyfasudil - a Rho kinase inhibitor - reduces these effects. Created with https://biorender.com/.

pericytes to regulate blood flow distribution within the renal medulla (*Pallone and Silldorff, 2001*; *Park et al., 1997*). In cerebral, retinal and cardiac pericytes, demonstrating pericyte α-SMA labelling has been difficult, but a more favourable fixative might increase the percentage of cells labelled (*Alarcon-Martinez et al., 2018*). In agreement with other studies, we observed that α-SMA protein was strongly labelled within the pericytes surrounding the descending vasa recta (*Figure 6k–n*; see also *Park et al., 1997*), including in pericytes near ischaemia-evoked blockage sites. Pericyte-specific deletion of α-SMA would allow assessment of whether this is the actin isoform conferring pericyte contractility (cf. *Alarcon-Martinez et al., 2019*). Furthermore, ischaemia increased MLC phosphorylation in pericytes (*Figure 6a–j*) and led to pericyte-mediated capillary constriction, consistent with actomyosin mediating the contractility of these cells.

Rho kinase, a key downstream effector of both endothelin-1 and angiotensin II, inhibits the MLC dephosphorylation required to relax pericytes (*Kimura et al., 1996*; *Maeda et al., 2003*), mainly by inhibiting eNOS activity (*Versteilen et al., 2006*),thus promoting constriction (*Hartmann et al., 2021*). Rho kinase also promotes actin polymerisation (*Kutcher and Herman, 2009*; *Maekawa et al., 1999*; *Zhang et al., 2018a*). We found that blocking Rho kinase with hydroxyfasudil reduced MLC phosphorylation in pericytes after ischaemia (*Figure 6a–j*), and reversed ischaemia-evoked pericyte-mediated

capillary constriction (hydroxyfasudil increased the capillary diameter specifically at pericyte somata (*Figure 4g*) in ischaemic animals, implying that the effects of Rho kinase inhibition were on renal pericytes rather than an extra-renal systemic action). This could explain why Rho kinase block reduces acute kidney injury (*Kentrup et al., 2011*; *Prakash et al., 2008*; *Teraishi et al., 2004*; *Versteilen et al., 2011*; *Versteilen et al., 2006*), as we have confirmed using kidney injury molecule-1 (Kim-1) as a marker (*Figure 7c and d*). In addition to inhibiting pericyte-mediated capillary constriction, hydroxyfasudil may also reduce kidney injury by reducing microvascular leukocyte accumulation, possibly by increasing the activity of eNOS (*Versteilen et al., 2011*; *Yamasowa et al., 2005*). It will be of interest to assess the efficacy of Rho kinase block for preventing kidney injury after longer periods of ischaemia than the 1 hr that we employed. In agreement with our findings in kidney pericytes, Rho kinase inhibition can block optogenetically induced constriction of brain capillaries by pericytes (*Hartmann et al., 2021*). Taken together, these findings support the concept that ischaemia-evoked capillary constriction reflects renal pericytes generating an actomyosin-dependent contraction, rather than there being a non-specific mechanism of constriction such as pericyte swelling.

Hydroxyfasudil is the active metabolite of fasudil, a drug that has been clinically approved in Japan since 1995 for the treatment of vasospasm following subarachnoid haemorrhage (*Lingor et al., 2019*). Fasudil treatment improves stroke outcome in animal models (*Vesterinen et al., 2013*) and humans (*Shibuya et al., 2005*) and our data suggest that it may also be useful for reducing post-ischaemic renal no-reflow and kidney damage.

We considered possible non-pericyte explanations for post-ischaemic capillary constriction and block. Post-ischaemic erythrocyte congestion in vasa recta has previously been described (*Crislip et al., 2017*; *Olof et al., 1991*) however physically adhering red blood cells do not physically cause the capillary blockages observed after ischaemia as they were associated with only a small percentage of block sites (*Figure 4—figure supplement 1a, b*). Thus, red blood cell trapping could be a consequence rather than a cause of the blockages. However, we cannot rule out the possibility that we observed only a small percentage of red blood cells in the capillary lumen because they are more readily displaced during the perfusion with PBS and PFA followed by FITC-albumin in gelatin than with protocols that do not perform transcardial perfusion or perform it only once (*Gaudin et al., 2014*; *Yemisci et al., 2009*). Leukocyte trapping may also contribute to reducing blood flow, but occurs on a longer time scale than we have studied (*Kelly et al., 1994*; *Rabb et al., 1995*; *Ysebaert et al., 2000*). Similarly, although a degradation of the eGCX has been reported after ischaemia (*Snoeijs et al., 2010*; *Song et al., 2018*), we found a uniform distribution of the eGCX along the vessel wall, which was not modified after ischaemia (*Figure 4—figure supplement 1e, h*), thus ruling out a causal association with capillary blockages which are preferentially located near pericytes. The present study demonstrates that pericyte-mediated constrictions of the descending vasa recta and cortical peritubular capillaries contribute to no-reflow and kidney injury at early stages of reperfusion; however, we cannot exclude the possibility that other factors, such as inflammation and leukocyte infiltration (*Gandolfo et al., 2009*; *Kelly et al., 1994*; *Rabb et al., 1995*; *Ysebaert et al., 2000*), or eGCX dysfunction (*Bongoni et al., 2019*), might also contribute to post-ischaemic microvascular injury at later phases of acute kidney injury. Furthermore, in response to the pericyte-mediated constriction evoked by ischaemia, the DVR may undergo post-ischaemic adaptations, releasing more nitric oxide at 48 hr post-ischaemia which could reduce pericyte constriction at later times after ischaemia than we have studied (*Zhang et al., 2018b*).

The recovery of blood flow in the medulla on renal arterial reperfusion was slower than in the cortex. The regulation of renal medullary blood flow is mainly mediated by vasa recta pericytes, independent of total or cortical blood flow (*Pallone and Silldorff, 2001*). The need for accurate flow regulation in the relatively hypoxic medulla may account for pericytes on the DVR being much closer together (mean separation 22.9 ± 0.9 μm) and with more circumferential processes (*Figure 4a–c*, *Figure 4—figure supplement 2b*) than for peritubular cortical pericytes (mean separation 41.3 ± 2.6 μm) which have mainly longitudinal (strand-like) processes (*Figure 5a–c*, *Figure 4—figure supplement 2a*) and

this may, in turn, contribute to a greater pericyte-mediated restriction of blood flow after ischaemia in the DVR than in the cortical capillaries. Despite these morphological differences between cortical and medullary pericytes, they showed similar immunoreactivity changes for p-MLC after ischaemia/ reperfusion and hydroxyfasudil treatment (*Figure 6a–j*). Perhaps surprisingly, given our data, in post-cadaveric renal transplants a better outcome has been reported for kidneys with a higher number of pericytes immediately post-transplant (*Kwon et al., 2008*). This may, however, reflect an aspect of pericyte function other than capillary constriction, such as angiogenesis and maintenance of vessel integrity (*Shaw et al., 2018*), with these functions failing in transplanted tissue in which pericytes have already died due to ischaemia.

In the brain, heart, and retina, contractile pericytes on capillaries play a key role in producing a prolonged reduction of blood flow after ischaemia (*Hall et al., 2014*; *O'Farrell et al., 2017*; *Yemisci et al., 2009*). Depending on the type of ischaemic model applied to these organs, collateral vessels may allow some (reduced) blood flow after the onset of ischaemia, which would lead to variable organ damage (*Farkas et al., 2007*; *Liu et al., 2019*; *Minhas et al., 2012*). In contrast, the kidney largely depends on the renal artery to provide a non-anastomotic supply to the glomeruli of each nephro-vascular unit (*Evans et al., 2013*; *Pallone et al., 2012*). Medullary hypoxia under normal conditions has been documented in several mammalian species, including humans (*Epstein et al., 1982*; *Leonhardt and Landes, 1963*). The medullary partial pressure of oxygen is ~10–20 mm Hg, contrasting with the partial pressure of oxygen in the cortex, which is ~50 mm Hg (*Brezis et al., 1994a*; *Brezis et al., 1991*; *Brezis et al., 1994b*). Thus, renal pericytes, especially in the medulla, are likely to be more susceptible to ischaemic injury than in other organs.

Rodent models of renal ischaemia can employ bilateral ischaemia or unilateral ischaemia with or without contralateral nephrectomy (*Fu et al., 2018*). In the present study, unilateral ischaemia without contralateral nephrectomy (which may occur during renal-sparing surgeries) (*Hollenbeck et al., 2006*; *Medina-Rico et al., 2017*) was chosen to explore the early mechanisms of ischaemia and reperfusion injury while using the contralateral kidney as a paired control for potential systemic haemodynamic changes that could be triggered during and after the surgical procedure. The presence of an unin-jured contralateral kidney reduces animal mortality during the surgical procedure, and thus longer ischaemia times can be used, resulting in more severe and reproducible injury (*Fu et al., 2018*; *Le Clef et al., 2016*; *Polichnowski et al., 2020*; *Soranno et al., 2019*). Unilateral ischaemia-reperfusion without contralateral nephrectomy is considered a strong model to study the progression from acute renal injury to long-term tubulo-interstitial fibrosis (*Fu et al., 2018*; *Le Clef et al., 2016*; *Polichnowski et al., 2020*; *Soranno et al., 2019*), but we acknowledge that the model used in the present study may not be similar to some clinical situations where both kidneys are injured, and there are limitations of translatability from all animal models of acute kidney injury to human disease (*Fu et al., 2018*). A limitation of our study is that all experiments were performed on male rats and mice. Female rats are relatively protected against post-ischaemic renal failure (*Lima-Posada et al., 2017*; *Müller et al., 2002*), possibly because in male rats androgens promote ischaemic kidney damage by triggering endothelin-induced vascular constriction (*Müller et al., 2002*). However, these studies showed that sex did not influence ischaemia repefusion-induced injury after 24 hr, but only after 7 days (*Lima-Posada et al., 2017*; *Müller et al., 2002*), which is on a much longer time scale than we have studied.

In the present study, we have shown that pericyte contraction contributes to reducing cortical and medullary blood flow at early stages of reperfusion. This initial pattern could also contribute to the pericyte injury, detachment and capillary rarefaction observed at later stages after ischaemia and reperfusion (*Kramann et al., 2017*), which lead to further damage to the kidney (*Khairoun et al., 2013*; *Kramann et al., 2017*). However, there was no evidence of pericyte detachment during the time frame of the present study. Treatment from the beginning of reperfusion (to mimic a clinically possible therapeutic approach) with hydroxyfasudil, a Rho kinase inhibitor, increased medullary and cortical blood flow, increased the post-ischaemic diameter of DVR capillaries at pericyte locations, reduced the percentage of DVR capillaries that remained blocked, and reduced kidney injury after renal reperfusion. Presumably, the protection of renal blood flow and downstream tissue health would be even greater if hydroxyfasudil could be given before ischaemia was induced (e.g. in situations such as cardiac surgery and kidney transplantation, where renal ischaemia might be anticipated). Thus, pericytes are a novel therapeutic target for reducing no-reflow after renal ischaemia. Acute kidney injury caused by post-ischaemic no-reflow causes significant socio-economic cost. Our identification

ofpericyte contraction as a therapeutic target for ischaemia-induced acute kidney injury should contribute to the development or re-purposing of drugs that can prevent renal no-reflow.

# Materials and methods

## Key resources table

| Reagent type (species) or resource | Designation | Source or reference | Identifiers | Additional information |
|---|---|---|---|---|
| Strain, strain background *Rattusnorvegicus* (Sprague Dawley, male) | Rat | UCL Biological Services | | |
| Genetic reagent (*Mus musculus/ spretus*, male) | NG2-DsRed mice | https://doi.org/10.1242/dev.004895 | JAX 008241 | |
| Antibody | anti-NG2 (mouse monoclonal) | AbCam | ab50009 | (1:200) |
| Antibody | Anti-Myosin light chain (phospho S20) (rabbit polyclonal) | AbCam | ab5694 | (1:100) |
| Antibody | kidney injury molecule-1 (Kim-1) (rabbit polyclonal) | NovusBiologicals | NBP1-76701 | (1:100) |
| Antibody | anti-alpha smooth muscle actin (rabbit polyclonal) | AbCam | ab5694 | (1:100) |
| Antibody | anti-glycophorin A (mouse monoclonal) | AbCam | ab9520 | (1:2000) |
| Antibody | Alexa Fluor 405 goat anti-rabbit (polyclonal) | ThermoFisher | A31556 | (1:500) |
| Antibody | Alexa Fluor 555 donkey anti-rabbit (polyclonal) | ThermoFisher | A31572 | (1:500) |
| Antibody | Alexa Fluor 555 donkey anti-mouse (polyclonal) | ThermoFisher | A31570 | (1:500) |
| Chemical compound, drug | isolectin B$_4$ - AlexaFluor 647 | ThermoFisher | I32450 | (1:200) |
| Chemical compound, drug | wheat germ agglutinin Alexa Fluor 647 conjugate | ThermoFisher | W32466 | 200 µl (1 mg/ml) |
| Chemical compound, drug | Hoechst 33,342 | ThermoFisher | H21492 | 1 mg/kg in 0.5 ml saline |
| Chemical compound, drug | gelatin | Sigma-Aldrich | G2625 | 5% in PBS |
| Chemical compound, drug | FITC-albumin | Sigma-Aldrich | A9771 | 1:200 in 5% gelatin |
| Chemical compound, drug | FITC-albumin | Sigma-Aldrich | A9771 | (1 mg in 100 µl; i.v.) |
| Chemical compound, drug | Hydroxyfasudilhydrochloride | Santa Cruz Biotechnology | sc-202176 | (3 mg/kg; i.v.) |
| Software, algorithm | *MATLAB R2015a* | MathWorks, Inc. | | in vivo data acquisition |
| Software, algorithm | ImageJ | https://imagej.nih.gov/ij/ | | imageanalysis |
| Software, algorithm | GraphPadPrism 6 | GraphPad Software, Inc | | statisticalanalysis |
| Other | DAPI stain | Molecular Probes | D1306 | 200 µl (5 µg/ml) |

## Study approval

Experiments were performed in accordance with European Commission Directive 2010/63/EU and the UK Animals (Scientific Procedures) Act (1986), with approval from the UCL Animal Welfare and Ethical Review Body.

## Animal preparation for ischaemia experiments

Due to the high density of kidney tissue, intravital microscopy is limited to superficial regions of the cortex <100 µm deep (*Sandoval and Molitoris, 2017*). As the renal medulla is inaccessible for in vivo imaging, we used laser Doppler flowmetry to assess blood flow changes of both kidneys or within the cortex and medulla of one kidney simultaneously. Additionally, we used FITC-albumin gelatin perfusion for measuring microvascular network perfusion (*O'Farrell et al., 2017*) in the renal cortex and medulla, supplemented with high-resolution images of individual capillaries to assess the mechanisms underlying blood flow changes.

Adult male Sprague-Dawley rats (P40-50), or NG2-dsRed male mice (P100-120) expressing dsRed in pericytes to allow live pericyte imaging, were anesthetized with pentobarbital sodium (induction 60 mg/kg i.p.; maintenance 10–15 mg/kg/h i.v.). The femoral veins were cannulated to administer anesthetic and drugs. Stable kidney perfusion was confirmed using laser Doppler probes (OxyFlo Pro 2-channel laser Doppler, Oxford, United Kingdom) to measure blood flow in the contralateral kidney throughout the experiment, and anesthesia was monitored by the absence of a withdrawal response to a paw pinch. Body temperature was maintained at 37.0°C ± 0.5°C with a heating pad.

## Renal ischaemia and reperfusion

Both kidneys were exposed, and the renal arteries and veins were dissected. Left kidneys were subjected to 60 min ischaemia by renal artery and vein cross-clamp, followed by 30 or 60 min reperfusion. This reperfusion duration was chosen to assess pericyte function soon after starting reperfusion. Right kidneys underwent the same procedures without vessel clamping. Two laser Doppler single-fibre implantable probes of 0.5 mm diameter (MSF100NX, Oxford Optronix, Oxford, United Kingdom) measured simultaneously the perfusion of both kidneys (or of the outer medulla and cortex of one kidney). Cortical and outer medullary perfusion were measured with the probe on or 2 mm below the kidney surface, respectively. Successful artery and vein occlusion was confirmed by a sudden fall of laser Doppler signal. Laser Doppler monitoring, which detects the movement of cells in the blood, is a widely used method for studies of microvascular perfusion in experimental and clinical studies and measures the total local microcirculatory blood perfusion in capillaries, arterioles, venules and shunting vessels (*Fredriksson et al., 2009*; *Rajan et al., 2008*). Laser Doppler is suitable for monitoring of relative renal microvascular blood flow changes in response to physiological and pharmacological stimuli in rodents (*Lu et al., 1993*; *Vassileva et al., 2003*).

Endothelial glycocalyx (eGCX) was labelled in vivo using wheat germ agglutinin (WGA) Alexa Fluor 647 conjugate (ThermoFisher, W32466, Waltham, MA) injected through the jugular vein (200 µl, 1 mg/ml) 45 min before renal ischaemia/reperfusion (*Kutuzov et al., 2018*). WGA binds to N-acetyl-D-glucosamine and sialic acid residues of the eGCX. Using ImageJ, WGA fluorescence intensities were measured by drawing regions of interest (ROIs) across capillaries at the mid-points of pericyte somata, and away from the soma in 5 µm increments on both sides of the pericyte. Capillary diameters were also measured at each position.

Hydroxyfasudil hydrochloride, a reversible cell-permeable inhibitor of Rho kinase (Santa Cruz Biotechnology sc-202176, Dallas, TX) which is expected to decrease pericyte contractility (*Hartmann et al., 2021*; *Kutcher et al., 2007*) was administered as a bolus (3 mg/kg *i.v.*), immediately on starting reperfusion. This protocol, rather than having the drug present during the ischaemic insult, better mimics a clinical situation where drugs could be given on reperfusion. Control and non-treated ischaemic animals received saline infusion with the same volume.

## Animal perfusion and tissue preparation for imaging

After renal ischaemia/reperfusion, animals were overdosed with pentobarbital sodium and transcardially-perfused with phosphate-buffered saline (PBS) (200 ml) followed by 4% paraformaldehyde (PFA, 200 ml) fixative and then 5% gelatin (20 ml in PBS Sigma-Aldrich, G2625, Darmstadt, Germany) solution containing FITC-albumin (Sigma-Aldrich, A9771, Darmstadt, Germany), followed

by immersion in ice for 30 min (adapted from *Blinder et al., 2013*). Kidneys were fixed overnight in 4% PFA, and 150 µm longitudinal sections made for immunohistochemistry. Rats have ~64 ml of blood per kg bodyweight, thus the FITC-albumin gelatin solution would suffice to fill the total blood volume. The gelatin sets when the body temperature falls and traps FITC-albumin in the perfused vessels; blocked vessels show no penetration of FITC-albumin past the block.

## In vivo two-photon imaging

NG2-DsRed mice (P100-120) were anesthetized using urethane (1.55 g/kg i.p., in two doses 15 min apart). Anesthesia was confirmed by the absence of a paw pinch withdrawal response. Body temperature was maintained at 36.8°C ± 0.3°C. A custom-built plate, attached to the kidney using superglue and agarose created a sealed well filled with phosphate-buffered saline during imaging, when the plate was secured under the objective on a custom-built stage.

Peritubular capillary diameter was recorded during renal ischaemia/reperfusion using two-photon microscopy of the intraluminal FITC-albumin (1 mg in 100 µl of saline given intravenously). Two-photon excitation used a Newport-Spectra Physics Mai Tai Ti:Sapphire Laser pulsing at 80 MHz, and a (Zeiss LSM710, Oberkochen, Germany) microscope with a 20 × water immersion objective (NA 1.0). Fluorescence was excited using 920 nm wavelength for DsRed, and 820 nm for FITC-albumin and Hoechst 33,342. Mean laser power under the objective was <35 mW. Images were analysed using ImageJ. Vessel diameter was defined using a line drawn across the vessel as the width of the intraluminal dye fluorescence.

## Immunohistochemistry

Pericytes were labelled by expression of DsRed under control of the NG2 promoter (in mice), or with antibodies to NG2 (1:200; Abcam ab50009, Cambridge, United Kingdom), α-smooth muscle actin (α-SMA) (1:100; Abcam ab5694, Cambridge, United Kingdom), or myosin light chain (phospho S20, 1:100, Abcam ab2480, Cambridge, United Kingdom), and the capillary basement membrane and pericytes were labelledwith isolectin B₄-Alexa Fluor 647 (1:200, overnight; Molecular Probes, I32450, Thermo Fisher Scientific, Waltham, MA). Z-stacks of the cortex and outer medulla (frame size 640.17 × 640.17 µm) for cell counting were acquired confocally (Zeiss LSM 700, Oberkochen, Germany). Pericyte intersoma distance was calculated between pairs of pericytes on capillaries within the same imaging plane. Kidney damage was assessed using kidney injury molecule-1 (Kim-1) antibody (1:100, overnight; Novus Biologicals, NBP1-76701, Abingdon, United Kingdom). Red blood cells were labelled with antibody to glycophorin A (1:2000, AbCam ab9520, Cambridge, United Kingdom). Alexa Fluor conjugated secondary antibodies were added overnight (1:500; ThermoFisher, A31572, A31556, A31570, Waltham, MA).

## Image analysis

Regions of interest (ROIs) were drawn around the renal cortex and medulla (*Figure 1*). The cortex thickness, which ranges from 1.5 mm to 3 mm in rodents, was defined as the distance from the renal surface (capsule) to the base of the medullary pyramid (*Andersen et al., 2020*; *Missbach-Guentner et al., 2018*; *Nogueira et al., 2016*). The FITC-albumin perfusion coupled with image threshold application also helped to visualise the cortical vessels and medullary rays in order to define the corticomedullary boundary.The mean FITC-albumin signal intensity was measured for each ROI using ImageJ. This signal is assumed to provide an approximate measure of the amount of blood perfusing the tissue (conceivably downstream capillary constriction could lead to an upstream dilation and an increased blood volume being detected but, if this did occur, it would lead to an underestimate of the decrease of perfusion occurring). To gain a more accurate assessment of perfusion, we also used the ImageJ macro TubeAnalyst (Advanced Digital Microscopy Core Facility at IRB Barcelona) to measure the microvascular network 'skeleton' of the renal cortex and medulla and obtain the total perfused capillary length, the number of perfused capillary segments and the overall perfused microvascular volume fraction (*Figure 2b–d*). To quantify the percentage of perfused capillaries, we counted the number of filled (with FITC-albumin) and unfilled vessels that crossed a line drawn through the centre of each image perpendicular to the main capillary axis.

To assess whether pericytes cause flow blockages, we measured the distance along the capillary from the termination of the FITC-albumin signal to the mid-point of the nearest visible pericyte

soma, since in brain most contractile circumferential pericyte processes (which can adjust capillary diameter) are near the pericyte soma (see *Figures 4d and 5f*, S2 and S3 of *Nortley et al., 2019*). Capillary diameters were measured at the block sites where the FITC-albumin signal terminated. We also plotted the diameter of the FITC-albumin labelled capillary lumen as a function of the distance from the pericyte somata to assess whether diameter reduction was a nonspecific effect of ischaemia, or was pericyte-related. A constriction seen specifically at pericyte somata is an unambiguous indication that pericyte contraction is occurring (*Nortley et al., 2019*). The identification, and direction of flow, of the afferent and efferent arterioles were deduced from tracking in confocal Z-stacks.

For quantification of the p-MLC levels in cortical and medullary pericytes, we selected regions of interest (ROIs) over pericytes after applying to maximum intensity projected stack images a lower and upper threshold, which was similar for all experimental groups (typically 50–150 in 8-bit images). Then, we used the ROIs thus selected to measure the mean fluorescence intensity over all the pericytes in each image. The background signal for each stack was obtained by placing a ROI in the parenchyma, away from but close to, the pericytes and the measured background fluorescence signal was subtracted from the mean intensity measured in the pericyte ROIs.

## Statistics

Statistical analysis employed Graphpad Prism (San Diego, CA). Data normality was tested with Shapiro-Wilk tests. Normally distributed data were compared using Student's 2-tailed t-tests or ANOVA tests. Data that were not normally distributed were analysed with Mann-Whitney or Kruskal-Wallis tests. *P* values were corrected for multiple comparisons using a procedure equivalent to the Holm-Bonferroni method or Dunn's test (corrected *P* values are significant if they are less than 0.05).

## Acknowledgements

We thank Jonathan Lezmy, Svetlana Mastitskaya and Thomas Pfeiffer for comments on the manuscript.

## Additional information

### Funding

| Funder | Grant reference number | Author |
| --- | --- | --- |
| Rosetrees Trust and Stoneygate Trust | | Felipe Freitas David Attwell |
| Wellcome Trust | | David Attwell |
| European Research Council | | David Attwell |

The funders had no role in study design, data collection and interpretation, or the decision to submit the work for publication.

### Author contributions

Felipe Freitas, Conceptualization, Data curation, Formal analysis, Investigation, Methodology, Supervision, Validation, Visualization, Writing – original draft, Writing – review and editing; David Attwell, Conceptualization, Funding acquisition, Project administration, Resources, Supervision, Writing – review and editing

### Author ORCIDs

Felipe Freitas http://orcid.org/0000-0003-4627-3509
David Attwell http://orcid.org/0000-0003-3618-0843

### Ethics

Experiments were performed under UK government Home Office licence 70/8976 in accordance with European Commission Directive 2010/63/EU and the UK Animals (Scientific Procedures) Act (1986), with approval from the UCL Animal Welfare and Ethical Review Body.

Decision letter and Author response
Decision letter https://doi.org/10.7554/eLife.74211.sa1
Author response https://doi.org/10.7554/eLife.74211.sa2

## Additional files

### Supplementary files
• Transparent reporting form

### Data availability
All data generated or analysed during this study are included in the manuscript and supporting file.

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
