## [Editor Report]

This paper identifies constriction of capillary pericytes as the underlying cause of post-ischemia renal no-reflow conditions, which contribute to kidney injury in a variety of settings, including cardiac surgery, renal transplantation and severe haemorrhage. As such, it should be of considerable interest to clinicians, but also to basic researchers in vascular biology, physiology and related fields. Data obtained strongly support a role for RhoA/Rho kinase in regulating the contractility of capillary pericytes.

---

## [Decision Letter]

**Decision letter after peer review:**

Thank you for submitting your article "Pericyte-mediated constriction of renal capillaries evokes no-reflow and kidney injury following ischemia" for consideration by *eLife*. Your article has been reviewed by 3 peer reviewers, including Mark T Nelson as Reviewing Editor and Reviewer #1, and the evaluation has been overseen by Martin Pollak as the Senior Editor. The following individual involved in review of your submission has agreed to reveal their identity: Turgay Dalkara (Reviewer #3).

Essential revisions:

1. Provide background information on pericyte contractile mechanisms in the Introduction. This should include consideration of the role of ROCK (both MLC- and endothelial cell eNOS/NO-based mechanisms) and α-SMA.

2. Elevate the impact of the paper and suitability for *eLife* by performing additional mechanistic studies:

a. Test whether the retinal α-SMA-dependent pericyte contraction mechanism reported by Arcon-Martinez et al., operates in the kidney.

b. Determine whether HF acts by inhibiting MLC dephosphorylation in pericytes, eNOS expression/NO activity in endothelial cells, or both, and whether the relative contribution of these mechanisms differs between the cortex and medulla.

3. Consider including a schematic depiction of the proposed model.

4. It is suggested to perform a long term experiment with assessment at 7 days from ischemia re perfusion and treatment with hydroxyfasudil.

5. It would be interesting to know the distribution of contracted pericyte profile in renal microcirculation; i.e. ensheathing type, junctional or strand-like and whether all exhibit p-MLC immunoreactivity, suggesting actomyosin contraction and, fasudil treatment differentially affects their contraction.

6. Another translationally important point could be to determine the therapeutic window of hydroxyfasudil, that is; if its beneficial effect is sustained after longer ischemia, which likely further increases pericyte contractions (Lee et al., Applied Optics, 55:9526-31, 2016). At least, this issue should be discussed. Moreover, comparison with the brain, heart and retina ischemia/no-reflow could provide some insight to assess if renal pericytes are particularly vulnerable to ischemia. Similarly, a brief comparative introduction to the renal microcirculation and capillary pericytes would be helpful for the reader.

7. Authors note that the vasculature was perfused with PBS to remove loose RBCs before perfusing PFA and FITC- albumin, so the only RBCs remaining should be those bound to the vessel walls. One would expect RBCs should be entrapped at the capillary segments where lumen was significantly narrowed by pericytes as shown in the brain (Yemisci et al., 2009; Gaudin et al., 2014). Lowering the transcardial perfusion pressure may disclose the RBC rouleaus trapped in microcirculation. This point should be taken into consideration in discussing the relevant findings.

8. Please give the rationale/reasons of having low number of rats in the medullary flow and ISCH^+^HF groups (n=4). Also, the stack numbers are smaller in ISCH-HF group compared to the other groups even after normalization for lower number of rats. Were these N values chosen based on power calculations?

9. Figure 1 c and d, ischemia label should read isch/rep (or I/R) in this and following figures because they were taken 30 minutes after reperfusion to indicate that the kidneys were already reperfused.

Please explain how the border of cortex and medulla are defined in non-ischemic kidney (C).

10. It looks like the shapes of the kidney sections illustrated are not from the same level so, the large vessels penetrating into the kidney cannot be seen in d and e, giving the impression that large vessels were not reperfused. Also, in E, only a few large vessels seem to be reperfused. The possibility that slow flow could also enhance fluorescence captured from large vessels should be considered.

11. Legend the last sentence could be read easier as "Statistical COMPARISONS used the number of animals as the N value NOT THE STACK NUMBERS".

12. Figure 2. Only after seeing Figure 5a, I could understand how renal microcirculation looks like. In a, the merged image is not helpful, instead a graphic illustration may provide better insight to renal circulation. In b-d, please explain how the skeleton was obtained and annotate them to indicate the arterioles and capillaries.

13. Figure 3. NG2 labeling seems to be visible only on large vessels at this low power magnification. Therefore, it may be better to use symbols differentiating arterioles and pericytes.

14. Figure 5. Was Hoechst applied epidurally? If so, please give reference showing that it does not affect cellular functions. In d, control capillary diameter is around 11 µm unlike in Figure 4g. Please explain. If pre-capillary arterioles were considered as capillaries, please indicate so for comparison with the data from other labs excluding the pre-capillary arterioles.

15. Figure 6d, is very convincing so, it might be a good idea to add a label on it, indicating that pericytes made visible with p-MLC although already given above on the right.

16. Suppl. Figure c, d. Please indicate the number of animals in each treatment group.

---

## [Author Response]

Essential revisions:1. Provide background information on pericyte contractile mechanisms in the Introduction. This should include consideration of the role of ROCK (both MLC- and endothelial cell eNOS/NO-based mechanisms) and α-SMA.

As requested we have now discussed the molecular basis of pericyte contractility and capillary constriction.

On page 3 we highlight the previously controversial role of α-SMA-based actomyosin-mediated contraction of pericytes which has been shown to mediate capillary constriction in the ischaemic retina. We now state (on page 3, last para – page 4, para 1):

“In the retina it has been shown that … capillary constriction is mediated by α-smooth muscle actin (α-SMA) based actomyosin-mediated contraction of capillary pericytes (Alarcon-Martinez et al., 2019). In the kidney, pericytes are associated with the cortical and medullary peritubular capillaries and the descending vasa recta. As in the retina, pericyte populations in the kidney, particularly those in the descending vasa recta, are associated with α-SMA expression and contractility (Park et al., 1997; Shaw et al., 2018).

On page 4 we have added a substantial section on the Rho kinase pathway and eNOS. This states:

“An important regulator of pericyte contractility is the Rho kinase pathway (Durham et al., 2014; Kutcher et al., 2007), which inhibits myosin phosphatase, thus increasing phosphorylation of MLC and increasing contraction (Kimura et al., 1996; Maeda et al., 2003). Overactivity of Rho kinase may play a key role in hypertension and diabetes, as well as in kidney ischemia (Jahani et al., 2018; Kushiyama et al., 2013; Peng et al., 2008; Soga et al., 2011; Versteilen et al., 2006). Rho kinase may also regulate pericyte contractility by modulating actin polymerization (Kureli et al., 2020; Kutcher and Herman, 2009; Maekawa et al., 1999; Zhang et al., 2018a). In ischaemia, an important pathway by which Rho kinase inhibits myosin phosphatase is via inactivation of endothelial nitric oxide synthase (eNOS) (Versteilen et al., 2006), thus reducing production of nitric oxide (NO). NO acts on guanylate cyclase to raise the concentration of cyclic GMP, which increases MLC phosphatase activity and thus decreases contraction, so inhibiting eNOS will increase MLC phosphorylation and contraction. Thus, both the direct effect of Rho kinase (Kimura et al., 1996) and its actions on eNOS (Versteilen et al., 2006) converge to promote MLC phosphorylation and contraction. Rho kinase is an important effector of vasoconstrictors such as endothelin-1 (Prakash et al., 2008; Wilhelm et al., 1999; Yamamoto et al., 2000) and angiotensin II (Rupérez et al., 2005), but its effects on pericytes are under-studied, although it may control their contractility (Durham et al., 2014; Hartmann et al., 2021; Homma et al., 2014; Kutcher et al., 2007; Pearson et al., 2013).”

2. Elevate the impact of the paper and suitability for eLife by performing additional mechanistic studies:a. Test whether the retinal α-SMA-dependent pericyte contraction mechanism reported by Arcon-Martinez et al., operates in the kidney.

We believe that the data already present in the paper demonstrate the involvement of α-SMA and actomyosin mediated contraction to a similar level of validation as the Alarcon-Martinez-papers, for the following reasons.

For the retinal pericytes, Alarcon-Martinez et al., (2018, 2019) indicated a role for α-SMA in mediating pericyte constriction in several ways. First, importantly, in their 2018 paper they demonstrated unequivocally that α-SMA is present in CNS pericytes (which had previously been controversial). Secondly, in 2019 they showed that inhibiting actin function using phalloidin, or by siRNA of α-SMA, inhibited ischaemia-evoked constriction. These data are certainly strongly suggestive of a crucial role for α-SMA, but a sceptic could point out that:

(i) phalloidin binds to all actin variants (not just α-SMA), so theoretically a different actin isoform could be involved, and

(ii) the knockdown of α-SMA in all cell types could have affected the morphology and spatial organisation of astrocytes and endothelial cells and thereby affected pericyte function.

For the kidney pericytes, we show that α-SMA protein was strongly labelled within the pericytes both surrounding the descending vasa recta and in the renal cortex (Figure 6d, i; see also Park et al., 1997), and that ischaemia greatly increased myosin light chain phosphorylation (a change known to induce smooth muscle contraction) in pericytes in both the medulla and the cortex, and that this was blocked by inhibiting the Rho kinase pathway (Figure 6e, j). Thus, our data confirm previous reports that pericytes in the kidney are associated with α-smooth muscle actin (α-SMA) expression and contractility (Shaw et al., 2018, Nat Rev Nephrol 14, 521; Park et al., 1997, Am J Physiol, 273, R1742) – indeed α-SMA is widely used as a marker for renal pericytes (Shaw et al., 2018, Nat Rev Nephrol 14, 521; Park et al., 1997, Am J Physiol, 273, R1742). Importantly, our documentation of changes in myosin light chain phosphorylation very strongly suggests that the pericyte contractility is mediated by actomyosin (rather than by a non-specific mechanism such as cell swelling).

We have now discussed the evidence supporting the idea that α-SMA-based actomyosin mediates contractility in pericytes on page 12 (para 3) to page 13 (para 1). We also explicitly state that pericyte-specific deletion of α-SMA might allow assessment of whether this is the actin isoform that confers contractility on pericytes on page 13, para 1.

b. Determine whether HF acts by inhibiting MLC dephosphorylation in pericytes, eNOS expression/NO activity in endothelial cells, or both, and whether the relative contribution of these mechanisms differs between the cortex and medulla.

There appears to be some misunderstanding in the referee’s question, as follows.

(1) First, hydroxyfasudil relaxes the pericytes (Figure 4g) and increases blood flow (Figure 1), which would imply that it promotes MLC dephosphorylation rather than inhibiting it.

(2) Second, in fact, we have already demonstrated in the paper that inhibition of Rho kinase with hydroxyfasudil (HF) increased MLC dephosphorylation in pericytes after ischaemia and reperfusion, as we show in Figure 6a-j.

(3) As detailed in the section described above (last para for point 1), which we have added to page 4:

“both the direct effect of Rho kinase (Kimura et al., 1996; Maeda et al., 2003) and its actions on eNOS (Versteilen et al., 2006) converge to promote MLC phosphorylation and contraction” so it is not meaningful to ask whether blocking Rho kinase acts by altering MLC phosphorylation or by altering eNOS activity, because the eNOS also ultimately acts by altering MLC phosphorylation.”

(4) It has already been shown by Versteilen et al., (2006, their Figure 2) that hydroxyfasudil’s protective effect on renal blood flow in ischaemia is very largely due to it decreasing eNOS activity. Since we have shown that the ischaemia/reperfusion-evoked reduction of renal blood flow is generated by pericytes (and not by arteriolar constriction) it is therefore expected that the effect of hydroxyfasudil on pericyte constriction will also be very largely as a result of an effect on eNOS activity.

To address the referee’s first question about eNOS, we have now:

(i) cited the Versteilen et al., (2006) paper on page 4 in the Introduction, where we also explain that the direct effect of Rho kinase and its effects via eNOS both converge on changes of MLC phosphorylation;

(ii) explained how the importance of eNOS could be confirmed by stating on page 11 (para 1) in the Results that:

“if pericyte contraction is via conventional smooth muscle actomyosin, the reduced MLC phosphorylation [in Figure 6] could explain the pericyte relaxation and increased blood flow evoked by Rho kinase inhibition. The data of Versteilen et al., (2006) suggest this is very largely mediated by inhibition of eNOS, which could be tested by quantifying the effect of eNOS block on the changes of MLC phosphorylation shown in Figure 6.”

To address the referee’s second question about pericytes in the medulla versus the cortex, we have now added extra data in response to the referee’s point 5 below. We show in the new Figure 4—figure supplement 2 that medullary and cortical pericytes tend to have a different morphology (with pericytes in the medulla having many more circumferential processes than those in the cortex). Furthermore, cortical pericytes are roughly twice as far apart as medullary pericytes. This implies that the same amount of contractile tone in medullary pericyte processes will produce more reduction of capillary diameter than it will in cortical pericytes. This has been explained on page 9, para 1, and on page 15, para 1.

3. Consider including a schematic depiction of the proposed model.

As requested, we have now added a schematic figure summarising the vasculature in the kidney and our findings on what happens to it in ischaemia, as a new Figure 8, and referred to it both at the start of the Introduction and at the start of the Discussion.

4. It is suggested to perform a long term experiment with assessment at 7 days from ischemia re perfusion and treatment with hydroxyfasudil.

Hydroxyfasudil evoked, not only an improvement of kidney perfusion, but also reduced kidney injury at 30 min after reperfusion following ischaemia, suggesting a critical role for Rho kinase in the development of renal reperfusion injury. Although we recognize that it would be interesting to carry out a long-term, translationally-oriented assessment of the effects of Rho kinase block on the outcome of kidney ischemia and reperfusion, we believe that these experiments are outside the scope of this study. The aim of the study was to investigate whether pericyte contraction contributes to reducing cortical and medullary blood flow early after ischaemia and reperfusion – an idea we have demonstrated is correct. While we expect therapeutic developments to result from this, carrying out such experiments would take a substantial investment of time, especially as our animal experimentation licence does not allow such “recovery” experiments (and it would take 6 months to obtain an amendment to the licence).

5. It would be interesting to know the distribution of contracted pericyte profile in renal microcirculation; i.e. ensheathing type, junctional or strand-like and whether all exhibit p-MLC immunoreactivity, suggesting actomyosin contraction and, fasudil treatment differentially affects their contraction.

We have now added, as Figure 4—figure supplement 2, representative images of pericytes on capillaries in the renal cortex and medulla. As noted above, and on page 9 (para 1) of the manuscript, cortical pericytes are roughly twice the distance apart that medullary pericytes are. The mean pericyte separation for medullary DVR capillaries (23 µm, see above) ensures that very few areas of the DVR are not associated with pericytes (Crawford et al., 2012, Nephron Physiol 120, p17). Furthermore cortical pericytes exhibit longitudinal processes but relatively few circumferential processes (thereby resembling strand-like pericytes on higher branch order capillaries in the brain), while medullary pericytes have a large number of circumferential processes (Figure 4—figure supplement 2b).

In the brain pericytes are associated both with straight capillary segments and with capillary branch points. In the kidney,the DVR are straight long capillaries that do not have many branching points (Figure 2b-d), so the branch point class of pericyte will be less common.

Despite these morphological differences between the cortical and medullary pericytes, previous researchers have shown that pericytes in both the cortex and medulla label for the contractile protein α-SMA, and we confirmed this for DVR pericytes near blockage sites (Figure 6k-n). In addition, both cortical and medullary pericytes displayed an increase in immunoreactivity for p-MLC after ischaemia and reperfusion (Figure 6). We also found that sites of ischaemia-evoked medullary and cortical capillary block were both associated with pericyte locations, and a reduced capillary diameter. The pericyte-mediated constriction and block of the descending vasa recta and cortical peritubular capillaries were also both reduced pharmacologically with hydroxyfasudil.

We have now stated all this on page 4 (para 1), page 9 (para 1) and page 15 (para 1).

6. Another translationally important point could be to determine the therapeutic window of hydroxyfasudil, that is; if its beneficial effect is sustained after longer ischemia, which likely further increases pericyte contractions (Lee et al., Applied Optics, 55:9526-31, 2016). At least, this issue should be discussed.

The Lee…Dalkara paper cited on the brain circulation shows that no-reflow is worse after a longer ischaemic period, but does not actually show any pericyte data, although this may well reflect more profound pericyte constriction – the only data we are are aware of showing the time course of pericyte-mediated capillary constriction is our own data on the brain microvasculature (Hall et al., 2014, Nature) but that employed chemical ischaemia as the insult which generates a more profound energy deprivation than real ischaemia. When tissue is made ischaemic it is a general finding that recovery is worse, the longer the ischaemia is applied for. We expect this to be the same for the kidney, but we have not carried out experiments to test it. We have now stated on page 13, para 2, that:

“It will be of interest to assess the efficacy of Rho kinase block for preventing kidney injury after longer periods of ischaemia than the one hour that we employed.”

Moreover, comparison with the brain, heart and retina ischemia/no-reflow could provide some insight to assess if renal pericytes are particularly vulnerable to ischemia.

In the brain, heart and retina, contractile pericytes on capillaries play a key role in reducing blood flow after ischemia (Yemisci et al., 2009; Hall et al., 2014; O'Farrell et al., 2017) because capillaries remain constricted by pericytes even when blood flow is restored to upstream arterioles. However, the sensitivity of pericytes in different organs to ischaemia, including, for example, whether they die in rigor, as found for profound (chemical) ischaemia in brain slices (Hall et al., 2014, Nature), will depend critically on the type of ischaemic model used and the possibility of a collateral circulation providing some nutrients (see e.g. Liu et al., 2019, J Am Heart Assoc, 8, e011220; Minhas et al., 2012, Front Neurol 11, 75; Farkas et al., 2007, Brain Res Rev 54, 162). In contrast, the kidney largely depends on the renal artery to provide a non-anastomotic supply to the glomeruli of each nephro-vascular unit (Pallone et al., 2012 Comp Physiol 2, 97; Evans et al., 2013 Clin Exp Pharmacol Physiol 40, 106). Medullary hypoxia under normal conditions has been documented in several mammalian species, including humans (Leonhardt et al., 1963, N Engl J Med 269,115; Epstein et al., 1982, Am J Physiol 243, F356-F363). The medullary partial pressure of oxygen is ~10-20 mm Hg, contrasting with the partial pressure of oxygen in the cortex, which is ~50 mm Hg (Brezis et al., 1991, J Clin Invest 88, 390; Brezis et al., 1994, Am J Physiol 267, F1059; Brezis et al., 1994 Am J Physiol 267, F1063). Thus, renal pericytes, especially in the medulla, are likely to be more susceptible to ischaemic injury than in other organs. As requested, we have now discussed all this on page 15 (para 2).

Similarly, a brief comparative introduction to the renal microcirculation and capillary pericytes would be helpful for the reader.

We have now added a diagram of the renal microcirculation as Figure 8. Page 3 (para 2) – page 4 (para 1) provides a brief introduction to the role of pericytes in kidney ischaemia from the perspective of what is known about the brain, retina and heart, and page 12 (para 3) – page 13 (para 1) gives a brief introduction to renal pericytes themselves.

7. Authors note that the vasculature was perfused with PBS to remove loose RBCs before perfusing PFA and FITC- albumin, so the only RBCs remaining should be those bound to the vessel walls. One would expect RBCs should be entrapped at the capillary segments where lumen was significantly narrowed by pericytes as shown in the brain (Yemisci et al., 2009; Gaudin et al., 2014). Lowering the transcardial perfusion pressure may disclose the RBC rouleaus trapped in microcirculation. This point should be taken into consideration in discussing the relevant findings.

We have now stated on page 14 (para 2) of the manuscript that:

“…we cannot rule out the possibility that we observed only a small percentage of red blood cells in the capillary lumen because they are more readily displaced during the perfusion with PBS and PFA followed by FITC-albumin in gelatin than with protocols that do not perform transcardial perfusion or perform it only once (Gaudin et al., 2014; Yemisci et al., 2009).”

8. Please give the rationale/reasons of having low number of rats in the medullary flow and ISCH^+^HF groups (n=4). Also, the stack numbers are smaller in ISCH-HF group compared to the other groups even after normalization for lower number of rats. Were these N values chosen based on power calculations?

Sample size estimates were not carried out prior to the study due to a lack of knowledge of effect sizes and response variance required for such calculations. Instead, once data were obtained with replicate numbers similar to those employed in previous studies, statistical analysis (reported in the paper) was used to assess the significance or otherwise of the results.

9. Figure 1 c and d, ischemia label should read isch/rep (or I/R) in this and following figures because they were taken 30 minutes after reperfusion to indicate that the kidneys were already reperfused.

As requested, we have now updated all the figure labels as suggested.

Please explain how the border of cortex and medulla are defined in non-ischemic kidney (C).

The cortex thickness, which ranges from 1.5 mm to 3 mm in rodents, was defined as the distance from the renal surface (capsule) to the base of the medullar pyramid (Nogueira et al., 2016, in vivo 30, 829; Missbach-Guentner et al., 2018, Scientific Reports 8, 1407; Andersen et al., 2020, Diagnostics (Basel) 10, 862). The FITC-albumin perfusion coupled with image threshold application also helped us to visualise the cortical vessels and medullary rays in order to define the corticomedullary boundary. This has now been stated on page 23, para 2.

10. It looks like the shapes of the kidney sections illustrated are not from the same level so, the large vessels penetrating into the kidney cannot be seen in d and e, giving the impression that large vessels were not reperfused. Also, in E, only a few large vessels seem to be reperfused. The possibility that slow flow could also enhance fluorescence captured from large vessels should be considered.

As requested, we have now updated the example images of the kidney sections in Figure 1.

11. Legend the last sentence could be read easier as "Statistical COMPARISONS used the number of animals as the N value NOT THE STACK NUMBERS".

As requested, we have now updated the figure legends as suggested.

12. Figure 2. Only after seeing Figure 5a, I could understand how renal microcirculation looks like. In a, the merged image is not helpful, instead a graphic illustration may provide better insight to renal circulation. In b-d, please explain how the skeleton was obtained and annotate them to indicate the arterioles and capillaries.

As requested, we have now added a schematic figure (Figure 8) depicting the renal microcirculation.

For Figure 2b-d we used the ImageJ macro TubeAnalyst (Advanced Digital Microscopy Core Facility at IRB Barcelona). The basic analysis implemented by TubeAnalyst includes segmentation, skeletonization and analysis of the perfused vasculature. The perfused microvessels are demarcated with an outline on the displayed image which dynamically updates its shape in response to adjustments done using the controls included in the analysis tab. Once the outline overlay matches the vessels in the displayed image, the analysis can be carried out. On completion of the analysis, the resulting image shows an overlay, which indicates the area encompassing all vessels, a skeletal representation of the vascular network and the computed branching points inside this area. We have not annotated the skeleton because: (i) the large number of vessels shown makes this almost impossible; and (ii) the great majority of the vessels in this region of the kidney are capillaries.

13. Figure 3. NG2 labeling seems to be visible only on large vessels at this low power magnification. Therefore, it may be better to use symbols differentiating arterioles and pericytes.

We have now edited the figure to facilitate the distinction between arterioles and pericytes.

14. Figure 5. Was Hoechst applied epidurally? If so, please give reference showing that it does not affect cellular functions.

Hoechst 33342 (1 mg/kg, in 0.5 ml of sterile, isotonic saline) was administered intravenously, as described by Dunn et al., 2018, Curr Protoc Cytom 83, 12.9.1. This paper has been cited in Figure 5 legend. It is impossible to be sure that Hoechst 33342 dos not affect any cell function, but it has been used to image live cells dividing, e.g. by Kukley et al., (2008, FASEB J, 22, 2957).

In d, control capillary diameter is around 11 µm unlike in Figure 4g. Please explain. If pre-capillary arterioles were considered as capillaries, please indicate so for comparison with the data from other labs excluding the pre-capillary arterioles.

In Figure 5d, the control capillary diameter of around 11 µm was acquired using 2-photon in vivo imaging of the mouse renal cortex microcirculation after intraluminal FITC-albumin was given intravenously. In contrast, the control diameter in Figure 4g was acquired in the descending vasa recta (DVR) in slices of rat renal medulla after perfusion with PBS/PFA and then FITC-albumin gelatin. Post-mortem effects, as well as the PFA perfusion-fixation in the medullary slices could explain the difference between these control values.

We did not consider pre-capillary arterioles as capillaries. Only peritubular capillaries were considered for our analysis.

15. Figure 6d, is very convincing so, it might be a good idea to add a label on it, indicating that pericytes made visible with p-MLC although already given above on the right.

We have now edited the figure, to indicate the pericytes labelled with p-MLC.

16. Suppl. Figure c, d. Please indicate the number of animals in each treatment group.

As requested, we have now updated the figure legends with the number of animals for each experimental group.